# Discrete-Convex-Analysis-Based Framework for Warm-Starting Algorithms with Predictions

**Shinsaku Sakaue**
The University of Tokyo
Tokyo, Japan
sakaue@mist.i.u-tokyo.ac.jp

**Taihei Oki**
The University of Tokyo
Tokyo, Japan
oki@mist.i.u-tokyo.ac.jp

## Abstract

Augmenting algorithms with learned predictions is a promising approach for going beyond worst-case bounds. Dinitz, Im, Lavastida, Moseley, and Vassilvitskii (2021) have demonstrated that warm-starts with learned dual solutions can improve the time complexity of the Hungarian method for weighted perfect bipartite matching. We extend and improve their framework in a principled manner via *discrete convex analysis* (DCA), a discrete analog of convex analysis. We show the usefulness of our DCA-based framework by applying it to weighted perfect bipartite matching, weighted matroid intersection, and discrete energy minimization for computer vision. Our DCA-based framework yields time complexity bounds that depend on the $\ell_\infty$-distance from a predicted solution to an optimal solution, which has two advantages relative to the previous $\ell_1$-distance-dependent bounds: time complexity bounds are smaller, and learning of predictions is more sample efficient. We also discuss whether to learn primal or dual solutions from the DCA perspective.

## 1   Introduction

Discrete optimization algorithms are applied to many real-world instances that take place repetitively. For example, recommendation systems repeat to solve bipartite matching instances to match users with services, and we solve a series of pixel-labeling instances to process images of a movie. Since such instances arising in the same domain often have some tendencies, using predictions made from past instances to improve algorithms' performance is a natural and promising idea. A recent line of work [36, 41, 4, 34, 43, 2, 5] successfully combined online algorithms with predictions and showed that those algorithms perform provably better than known worst-case bounds if predictions are good while enjoying worst-case guarantees even if predictions are poor. See [37] for a survey.

Dinitz et al. [17] recently initiated the study of improving the time complexity of algorithms with predictions. They focused on warm-starting the well-known Hungarian method for the weighted perfect bipartite matching problem with predictions on dual solutions, and obtained the time complexity of $\mathrm{O}(\min\{m\sqrt{n}\|\hat{p} - p^*\|_1, mn\})$, where $n$ and $m$ are the number of vertices and edges, respectively, and $\hat{p} \in \mathbb{R}^n$ is a prediction on an optimal dual solution $p^* \in \mathbb{R}^n$. That is, while the Hungarian method takes $\mathrm{O}(mn)$ time in the worst case, it can run faster given a good prediction. Dinitz et al. [17] also presented an algorithm for converting infeasible learned dual solutions into initial feasible solutions, and proved an $\mathrm{O}(C^2 n^3 \log n)$ sample complexity bound for learning $\hat{p}$ that approximately minimizes the expected $\ell_1$-error $\mathbb{E}\|\hat{p} - p^*\|_1$, assuming an optimal prediction is contained in $[-C, +C]^n$. They thus established an end-to-end framework for warm-starting the Hungarian method with predictions.

While Dinitz et al. [17] has shown that their prediction-based warm-start framework is effective for bipartite matching and $b$-matching, its application to other problems remains to be studied. Since their idea has the potential to yield strong *beyond-the-worst-case* time complexity bounds, the next question of theoretical interest is: *when and how can we warm-start algorithms with predictions?*

36th Conference on Neural Information Processing Systems (NeurIPS 2022).

Table 1: Our results for weighted perfect bipartite matching (BM) on bipartite graphs with $n$ vertices and at most $m$ edges, weighted matroid intersection (MI) on pairs of rank-$r$ matroids on an identical ground set of size $n$ ($\tau$ is the running time of independence oracles), and discrete energy minimization on graphs with $n$ vertices and at most $m$ edges.

| Problem | Local optimization problem | Time complexity | Prediction |
|---|---|---|---|
| Weighted perfect BM | Maximum cardinality BM | $\mathrm{O}(m\sqrt{n}\|p^* - \hat{p}\|_\infty)$ | Dual |
| Weighted MI | Maximum cardinality MI | $\mathrm{O}(\tau n r^{1.5}\|p^* - \hat{p}\|_\infty)$ | Dual |
| Discrete energy min. | Minimum cut | $\mathrm{O}(mn^2\|p^* - \hat{p}\|_\infty)$ | Primal |

**Our contribution** is to extend and improve the framework of [17] in a principled manner. Our idea comes from an intuition that the time complexity bound of [17] seems to be originating from some geometric property of the Hungarian method. In continuous optimization, warm-starting the gradient descent method reduces its running time, which we can see by simple geometric reasoning (see Figure 1a). We formalize this idea for discrete optimization via *discrete convex analysis* (DCA) by Murota [38], which is a discrete analog of convex analysis [42] and offers a gradient-descent-like interpretation of various discrete optimization algorithms, as in Figure 1b. Based on DCA, we show that warm-starting with predictions is effective for a large class of problems called *L/L♮-convex minimization*.

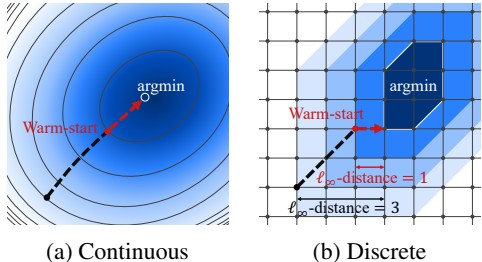

(a) Continuous      (b) Discrete

Figure 1: Images of warm-starts in (a) continuous and (b) discrete optimization, where darker colors indicate smaller objective values.

As with the framework of [17], given a prediction $\hat{p}$ on an optimal solution $p^*$, we convert $\hat{p}$ into an initial feasible solution $p^\circ$. Then, starting from $p^\circ$, we iteratively solve a local optimization problem to find a direction, along which we proceed. We will see that the number of iterations is $\mathrm{O}(\|p^* - \hat{p}\|_\infty)$ (see Figure 1b for an image). Thus, if a local optimization solver runs in $T_{\mathrm{loc}}$ time, the total time complexity is $\mathrm{O}(T_{\mathrm{loc}}\|p^* - \hat{p}\|_\infty)$ plus the time of converting $\hat{p}$ into $p^\circ$, which is often shorter than $T_{\mathrm{loc}}$. Table 1 summarizes the results obtained by applying our DCA-based framework to specific problems, where $T_{\mathrm{loc}}$ is replaced with the running time of standard local optimization solvers. As for bipartite matching, our bound is up to $n$ times smaller than the $\mathrm{O}(m\sqrt{n}\|\hat{p} - p^*\|_1)$ bound of [17].[1]

We then provide an $\mathrm{O}(C^2 n)$ sample complexity bound for learning predictions that approximately minimize the expected $\ell_\infty$-error, $\mathbb{E}\|p^* - \hat{p}\|_\infty$, assuming an optimal prediction to be in $[-C, +C]^n$. Our bound is better than the previous $\mathrm{O}(C^2 n^3 \log n)$ bound of [17] for approximately minimizing the expected $\ell_1$-error. Our method for learning predictions is based on a recent online-learning framework by Khodak et al. [31], and we also obtain an $\mathrm{O}(C\sqrt{nT})$ regret bound for the online setting.

We finally discuss whether to learn primal or dual solutions, which depends on problems as in Table 1 and has been decided in a somewhat ad-hoc manner in literature. We provide a guideline for choosing primal or dual from the DCA perspective by considering *path connectivity* of feasible regions.

## 1.1 Related work

**Theoretically fast algorithms.** We overview existing theoretically fast algorithms. We emphasize that, as with [17], our motivation is to accelerate simple algorithms, not to develop theoretically fast algorithms, which are often difficult to implement and empirically slow due to hidden large constants. For maximum cardinality bipartite matching, the standard Hopcroft–Karp algorithm [26] runs in $\mathrm{O}(m\sqrt{n})$ time, and a recent fast algorithm [47] takes $\mathrm{O}(m + n^{1.5})$ time (up to logarithmic factors). For weighted bipartite matching with non-negative integer weights bounded by $W$, a scaling-type algorithm [18] runs in $\mathrm{O}(m\sqrt{n}\log(W))$ time. Moreover, a recent almost-linear-time max/min-cost

---

[1]Unlike [17], our framework cannot yield worst-case bounds in general, which is a limitation of our work. This, however, does not matter in practice if we can run standard algorithms in parallel, as discussed in Section 6.

flow algorithm [14] implies an $O(m^{1+o(1)} \log^2 W)$-time algorithm for weighted bipartite matching. For maximum cardinality matroid intersection, a standard algorithm is Cunningham's $O(\tau n r^{1.5})$-time algorithm [15], and a recent faster algorithm [12] runs in $\tilde{O}(\tau n r \log r)$ time. For weighted matroid intersection, there is an $O(\tau n r^2)$-time algorithm [9]. Moreover, if the maximum weight is bounded by $W$, there are $O(\tau n^2 \log(nW))$-time [35] and $O(\tau n r^{1.5} W)$-time [27] algorithms. For discrete energy minimization, there is an $O(mn \log(n^2/m) \log(nW))$-time algorithm [1], where $W$ is the number of possible vertex labels. For a particular case where smoothness of vertex labels is measured by linear deviation functions, there is an algorithm that takes almost the same time to solve a min-cut instance [25]; combined with the algorithm of [14], it runs in $O(m^{1+o(1)} \log^2 W)$ time.

**Algorithms with predictions.** Many recent studies [36, 41, 4, 34, 43, 2, 5] improved competitive ratios of online algorithms with predictions. Dinitz et al. [17] proposed to warm-start algorithms with predictions. Khodak et al. [31] developed an online-learning framework for learning predictions, and applied it to the $\ell_1$-error minimization setting of [17] to obtain $O(Cn\sqrt{T})$ regret and $O(C^2 n^2)$ sample complexity bounds. By contrast, we learn predictions to minimize the $\ell_\infty$-error, yielding better guarantees. *Data-driven algorithm design* [3] is closely related to algorithms with predictions. As discussed in [31], one distinction is that the former aims to tune algorithm parameters to optimize the expected performance, while the latter focuses on prediction-dependent bounds to quantify the improvement gained by using predictions. A very recent study [13] also extended and improved [17], which should be considered as independent of each other and employs a different approach from ours.

## 2 Preliminaries

Let $\lfloor \cdot \rceil$ denote the rounding to the nearest integer (0.5 is rounded down). We apply $\lfloor \cdot \rfloor$, $\lceil \cdot \rceil$, and $\lfloor \cdot \rceil$ to vectors in an element-wise manner. We use $V = \{1, 2, \ldots, n\}$ to denote a finite ground set of size $n \in \mathbb{N}$. For any $X \subseteq V$, $\mathbf{1}_X \in \{0, 1\}^V$ denotes an indicator vector whose entries corresponding to $X$ are one and the others are zero; let $\mathbf{1} = \mathbf{1}_V$. Let $\vee$ and $\wedge$ denote the element-wise maximum and minimum operators, respectively. For any $S \subseteq \mathbb{Z}^V$, let $\mathrm{conv}(S) \subseteq \mathbb{R}^V$ denote the convex hull of $S$.

Given any function $g : \mathbb{Z}^V \to \mathbb{R} \cup \{+\infty\}$, let $\mathrm{dom}\, g = \left\{ p \in \mathbb{Z}^V \mid g(p) < +\infty \right\}$ be its *effective domain*, which indicates the feasible region of a minimization problem of form $\min_{p \in \mathbb{Z}^V} g(p)$. We say $g$ is *proper* if $\mathrm{dom}\, g \neq \emptyset$. In this paper, we assume the following basic conditions to hold.

**Assumption 1.** *We assume that any objective function $g : \mathbb{Z}^V \to \mathbb{R} \cup \{+\infty\}$ is proper and has at least one minimizer. Moreover, given any $g$, we uniquely associate $p^*(g) \in \mathrm{argmin}_{p \in \mathbb{Z}^V} g(p)$ with $g$ by breaking ties with an arbitrary predefined rule (to deal with the case of multiple minimizers).*

### 2.1 Background on discrete convex analysis

We overview the basics of discrete convex analysis. We refer the reader to [38] for more information.

Considering functions on $\mathbb{Z}^V$, how to define convexity is already nontrivial. A well-known property of a continuous convex function $f$ is midpoint convexity, i.e., $\frac{f(x)+f(y)}{2} \geq f\left(\frac{x+y}{2}\right)$, and its natural discrete analog for a function $g : \mathbb{Z}^V \to \mathbb{R} \cup \{+\infty\}$ would be $g(p) + g(q) \geq g\left(\left\lceil \frac{p+q}{2} \right\rceil\right) + g\left(\left\lfloor \frac{p+q}{2} \right\rfloor\right)$ for all $p, q \in \mathbb{Z}^V$. This condition is indeed equivalent to the following $L^\natural$-*convexity* of $g$.

**Definition 1.** A proper function $g : \mathbb{Z}^V \to \mathbb{R} \cup \{+\infty\}$ is *L-convex* if it has the following properties.

**Submodularity:** $g(p) + g(q) \geq g(p \vee q) + g(p \wedge q)$ for all $p, q \in \mathbb{Z}^V$.

**Linearity in the direction of 1:** there exists $r \in \mathbb{R}$ such that $g(p + \mathbf{1}) = g(q) + r$ for all $p \in \mathbb{Z}^V$.

A proper function $g : \mathbb{Z}^V \to \mathbb{R} \cup \{\infty\}$ is $L^\natural$-*convex* if a function $\tilde{g} : \mathbb{Z} \times \mathbb{Z}^V \to \mathbb{R} \cup \{+\infty\}$ defined by $\tilde{g}(p_0, p) = g(p - p_0 \mathbf{1})$ for all $p_0 \in \mathbb{Z}$ and $p \in \mathbb{Z}^V$ is L-convex.

While $L^\natural$-convex functions form a wider class than L-convex functions, the two classes are essentially equivalent due to the one-to-one correspondence between $L^\natural$-convex functions on $\mathbb{Z}^V$ and L-convex functions on $\mathbb{Z} \times \mathbb{Z}^V$. Thus, we can choose whichever is more convenient for modeling problems. Note that the sum of two $L/L^\natural$-convex functions, $g_1$ and $g_2$, is $L/L^\natural$-convex if it is proper.

---
**Algorithm 1** Steepest descent method for L-convex ($L^\natural$-convex) function minimization
---
1: $p \leftarrow p^\circ \in \operatorname{dom} g$              $\triangleright p^\circ$ is an initial feasible solution.
2: **while** not converged **:**
3:      $d \leftarrow \operatorname{argmin}\{ g(p + d') \mid d' \in \mathcal{N}_+ \}$       $\triangleright$ Replace $\mathcal{N}_+$ with $\mathcal{N}_\pm$ if $g$ is $L^\natural$-convex.
4:      **if** $g(p + d) = g(p)$ **:**
5:          **return** $p$
6:      $\lambda \leftarrow \sup\{ \lambda' \in \mathbb{Z}_{>0} \mid g(p + \lambda' d) - g(p) = \lambda'(g(p + d) - g(p))\}$   $\triangleright$ Alternatively, $\lambda \leftarrow 1$.
7:      $p \leftarrow p + \lambda d$
---

We say a non-empty set $S \subseteq \mathbb{Z}^V$ is $L/L^\natural$-convex if its indicator function $\delta_S$ is $L/L^\natural$-convex, where $\delta_S(p) = 0$ if $p \in S$ and $+\infty$ otherwise. If $g : \mathbb{Z}^V \to \mathbb{R} \cup \{+\infty\}$ is $L/L^\natural$-convex, then $\operatorname{dom} g \subseteq \mathbb{Z}^V$ is an $L/L^\natural$-convex set. The next proposition provides useful representations of $L/L^\natural$-convex sets.

**Proposition 1.** *A non-empty set $S \subseteq \mathbb{Z}^V$ is $L^\natural$-convex if and only if there exist $\alpha_i \in \mathbb{Z} \cup \{-\infty\}$, $\beta_i \in \mathbb{Z} \cup \{+\infty\}$, and $\gamma_{ij} \in \mathbb{Z} \cup \{+\infty\}$ ($i, j \in V; i \neq j$) such that*

$$S = \big\{ p \in \mathbb{Z}^V \mid \alpha_i \leq p_i \leq \beta_i, \ \ p_j - p_i \leq \gamma_{ij} \ \text{for } i, j \in V; i \neq j \big\},$$

*and is L-convex if and only if $S$ is written as above without the box constraints $\alpha_i \leq p_i \leq \beta_i$ ($i \in V$).*

Therefore, the colored area in Figure 1b illustrates an example of $L^\natural$-convex sets in $\mathbb{Z}^2$. We can also represent $\operatorname{conv}(S) \subseteq \mathbb{R}^V$ as above by replacing $\mathbb{Z}^V$ with $\mathbb{R}^V$ (see [38, Section 5.5]). $L/L^\natural$-convex sets also have the following useful property, which we prove in Appendix A.

**Lemma 1.** *Let $S \subseteq \mathbb{Z}^V$ be an $L/L^\natural$-convex set and $p \in \operatorname{conv}(S)$. Then, it holds $\lfloor p \rfloor \in S$.*

## 2.2 Steepest descent method for $L/L^\natural$-convex function minimization

An essential property of continuous convex functions is that the local optimality implies the global optimality, which enables the gradient descent method to reach a global optimum. L- and $L^\natural$-convexity inherits this property with respect to neighborhood $\mathcal{N}_+ = \{0, +1\}^V$ and $\mathcal{N}_\pm = \{0, -1\}^V \cup \{0, +1\}^V$, respectively, i.e., for any L-convex ($L^\natural$-convex) function $g : \mathbb{Z}^V \to \mathbb{R} \cup \{+\infty\}$ and $p \in \mathbb{Z}^V$, it holds

$$p \in \operatorname{argmin}\big\{ g(q) \mid q \in \mathbb{Z}^V \big\} \quad \Longleftrightarrow \quad g(p) \leq g(p + d) \quad \text{for every } d \in \mathcal{N}_+ \ (d \in \mathcal{N}_\pm).$$

This fact underpins the convergence of a steepest descent method (Algorithm 1) to a global minimum of any $L/L^\natural$-convex function $g : \mathbb{Z}^V \to \mathbb{R} \cup \{+\infty\}$, as stated in Proposition 2 presented below.

Algorithm 1 is very simple: starting from an initial point $p^\circ \in \operatorname{dom} g$, it finds a steepest direction by solving a local optimization problem (Step 3), sets the step length $\lambda$, and updates the solution (Algorithm 1 is a *long-step* version [24, 46], and we can also let $\lambda = 1$ in Step 6). The algorithm terminates if Step 3 does not improve the objective value. The local optimization problem in Step 3 can be written as $\min_{X \subseteq V} f(X) = g(p + \mathbf{1}_X)$ (or $g(p \pm \mathbf{1}_X)$ if $g$ is $L^\natural$-convex). From $L/L^\natural$-convexity of $g$, one can confirm that $f : 2^V \to \mathbb{R} \cup \{+\infty\}$ is a submodular function, which can be minimized in polynomial time in general, and more efficient local optimization solvers are available for many specific problems. Algorithm 1 thus provides an efficient way to minimize $L/L^\natural$-convex functions.

The number of iterations of Algorithm 1 is known to be bounded by the distance between an initial point and a global optimal solution. To describe this claim precisely, for any $p \in \mathbb{R}^V$, we define

$$\|p\|_\infty^\pm = \|p\|_\infty^+ + \|p\|_\infty^- \quad \text{where} \quad \|p\|_\infty^+ = \max_{i \in V} \max\{0, +p_i\} \quad \text{and} \quad \|p\|_\infty^- = \max_{i \in V} \max\{0, -p_i\}.$$

Note that $\|p\|_\infty \leq \|p\|_\infty^\pm \leq 2\|p\|_\infty$ holds, i.e., $\|p\|_\infty^\pm = \Theta(\|p\|_\infty)$. Moreover, $\|\cdot\|_\infty^\pm$ satisfies the axioms of norms, and hence we sometimes refer to it as the $\ell_\infty^\pm$-norm.

**Proposition 2** ([39, Theorem 1.2] and [24, Theorem 6.2]). *Algorithm 1 returns a global optimal solution to $\min_{p \in \mathbb{Z}^V} g(p)$ in at most $\|p^*(g) - p^\circ\|_\infty^\pm + 1 = O(\|p^*(g) - p^\circ\|_\infty)$ iterations.*

Note that Proposition 2 holds regardless of $L/L^\natural$ and the choice of $\lambda$ in Step 6. The original results shown in [39, 24] provide stronger bounds on the number of iterations by using a global optimal solution $p^*$ with the smallest $\|p^*(g) - p^\circ\|_\infty^\pm$ value, and we obtain Proposition 2 by replacing $p^*$ with an optimal solution $p^*(g)$ that is uniquely associated with $g$ by Assumption 1; we do this to get simple convex loss functions with which we can learn predictions (see Section 4).

# 3 DCA-based framework and its applications

We present our general DCA-based framework for L/L$^\natural$-convex minimization. Our basic idea is to warm-start Algorithm 1 with predictions and bound the number of iterations by using Proposition 2. Algorithm 1 has two parts that take considerable time: obtaining an initial feasible solution in Step 1 and solving a local optimization problem in Step 3. For now, we suppose that oracles for the two steps are available and present our main theorem, which, together with learning guarantees in Section 4, formalizes our DCA-based framework.

**Theorem 1.** *Let $g : \mathbb{Z}^V \to \mathbb{R} \cup \{+\infty\}$ be an L/L$^\natural$-convex function. If, for any $q \in \mathbb{R}^V$, we can compute an $\ell_\infty^\pm$-projection $\mathcal{P}(q) \in \arg\min\{\, \|p - q\|_\infty^\pm \mid p \in \mathrm{conv}(\mathrm{dom}\, g) \,\}$ in $T_{\mathrm{prj}}$ time and we can solve a local optimization problem in Step 3 in $T_{\mathrm{loc}}$ time, the following guarantees hold.*

1. *Given a possibly infeasible prediction $\hat{p} \in \mathbb{R}^V$, we can obtain an initial feasible solution $p^\circ = \lfloor \mathcal{P}(\hat{p}) \rceil \in \mathrm{dom}\, g$ in $\mathrm{O}(T_{\mathrm{prj}} + |V|)$ time.*

2. *Given the initial feasible solution $p^\circ = \lfloor \mathcal{P}(\hat{p}) \rceil$, Algorithm 1 computes an optimal solution to $\min_{p \in \mathbb{Z}^V} g(p)$ in $\mathrm{O}(T_{\mathrm{loc}} \| p^*(g) - \hat{p} \|_\infty)$ time.*

*Proof.* Regarding the first claim, the time complexity follows from the $T_{\mathrm{prj}}$-time projection and the $\mathrm{O}(|V|)$-time rounding, and $\lfloor \mathcal{P}(\hat{p}) \rceil \in \mathrm{dom}\, g$ follows from $\mathcal{P}(\hat{p}) \in \mathrm{conv}(\mathrm{dom}\, g)$ and Lemma 1. We below prove the second claim. Proposition 2 implies that we can compute an optimal solution in $T_{\mathrm{loc}}(\|p^*(g) - p^\circ\|_\infty^\pm + 1)$ time. We further bound $\|p^*(g) - p^\circ\|_\infty^\pm$ as follows. Since the rounding changes each entry up to $\pm 1/2$, $\|p^*(g) - p^\circ\|_\infty^\pm \le \|p^*(g) - \mathcal{P}(\hat{p})\|_\infty^\pm + 1$ holds. Furthermore, the triangle inequality of $\|\cdot\|_\infty^\pm$, $\mathcal{P}(\hat{p}) \in \arg\min_{p \in \mathrm{conv}(\mathrm{dom}\, g)} \|p - \hat{p}\|_\infty^\pm$, and $p^*(g) \in \mathrm{conv}(\mathrm{dom}\, g)$ imply $\|p^*(g) - \mathcal{P}(\hat{p})\|_\infty^\pm \le \|p^*(g) - \hat{p}\|_\infty^\pm + \|\mathcal{P}(\hat{p}) - \hat{p}\|_\infty^\pm \le 2\|p^*(g) - \hat{p}\|_\infty^\pm \le 4\|p^*(g) - \hat{p}\|_\infty$. Thus, the time complexity is $\mathrm{O}(T_{\mathrm{loc}} \|p^*(g) - \hat{p}\|_\infty)$, as desired. □

That is, given a prediction $\hat{p} \in \mathbb{R}^V$, we can solve $\min_{p \in \mathbb{Z}^V} g(p)$ in $\mathrm{O}(T_{\mathrm{prj}} + |V| + T_{\mathrm{loc}} \|p^*(g) - \hat{p}\|_\infty)$ time. We below discuss how large $T_{\mathrm{prj}}$ and $T_{\mathrm{loc}}$ can be for bipartite matching, matroid intersection, and discrete energy minimization; we also mention general L$^\natural$-convex function minimization. In all the cases, it turns out $T_{\mathrm{prj}} + |V| \le T_{\mathrm{loc}}$, implying the total time complexity of $\mathrm{O}(T_{\mathrm{loc}} \|p^*(g) - \hat{p}\|_\infty)$. Thus, replacing $T_{\mathrm{loc}}$ with those of standard local optimization solvers, we obtain the results in Table 1.

## 3.1 Weighted perfect bipartite matching

We consider the weighted perfect bipartite matching problem studied in [17]. Let $G = (V, E)$ be a bipartite graph with bipartition $V = L \cup R$, $|L| = |R| = n/2$ (where $n$ is even), and $|E| \le m$. Let $w \in \mathbb{Z}^E$ be edge weights. We assume that $V$ is fixed and $G$ has at least one perfect matching (which we can check by solving the maximum cardinality bipartite matching problem once). Under this condition, we allow both $w$ and $E$ to change over instances generated randomly or adversarially, as described in Section 4; this slightly extends the setting of [17], which fixes $E$.[2] The dual problem of the maximum weight perfect bipartite matching problem on $G$ is given as follows:[3]

$$\underset{s \in \mathbb{Z}^L, t \in \mathbb{Z}^R}{\text{minimize}} \quad \sum_{i \in L} s_i - \sum_{j \in R} t_j \quad \text{subject to} \quad s_i - t_j \ge w_{ij} \quad \forall (i, j) \in E. \tag{1}$$

We below use $p = (s, t) \in \mathbb{Z}^L \times \mathbb{Z}^R = \mathbb{Z}^V$ to denote the dual variables. The objective function is linear (L-convex), and the inequalities defining the feasible region can be written as in Proposition 1, whose indicator function is L-convex. Thus, if we let $g : \mathbb{Z}^V \to \mathbb{R} \cup \{+\infty\}$ be the sum of these two functions, we can write (1) as an L-convex function minimization problem of form $\min_{p \in \mathbb{Z}^V} g(p)$.

**Projection.** Given a prediction $\hat{p} = (\hat{s}, \hat{t}) \in \mathbb{R}^L \times \mathbb{R}^R$, we compute $\varepsilon = \max_{(i,j) \in E}(w_{ij} - \hat{s}_i + \hat{t}_j)$. If $\varepsilon \le 0$, $\hat{p}$ is already in $\mathrm{conv}(\mathrm{dom}\, g)$; otherwise, $(\hat{s} + \frac{\varepsilon}{2}\mathbf{1}, \hat{t} - \frac{\varepsilon}{2}\mathbf{1})$ gives an $\ell_\infty^\pm$-projection $\mathcal{P}(\hat{p})$, as in the following Lemma 2. The total computation time of this projection step is $T_{\mathrm{prj}} = \mathrm{O}(m)$.

---

[2] While the minimum-weight setting is studied in [17], we consider the maximum-weight setting for convenience. Note that we can deal with the minimum-weight setting since $w$ is allowed to have negative entries.

[3] Since edge weights $w$ are integer and the constraint is given by a totally unimodular matrix, there is at least one integral optimal solution. Hence, we can restrict the domain to $\mathbb{Z}^L \times \mathbb{Z}^R$.

**Lemma 2.** *For any $\hat{p} = (\hat{s}, \hat{t}) \in \mathbb{R}^L \times \mathbb{R}^R$ such that $\varepsilon > 0$, $\left(\hat{s} + \frac{\varepsilon}{2}\mathbf{1}, \hat{t} - \frac{\varepsilon}{2}\mathbf{1}\right)$ gives an $\ell^{\pm}_{\infty}$-projection of $\hat{p}$ onto $\mathrm{conv}(\mathrm{dom}\, g)$, the convex hull of the feasible region of* (1).

*Proof.* For any $\Delta s \in \mathbb{R}^L$ and $\Delta t \in \mathbb{R}^R$, we have $(\hat{s} + \Delta s, \hat{t} + \Delta t) \in \mathrm{conv}(\mathrm{dom}\, g)$ if and only if $\Delta s_i - \Delta t_j \geq w_{ij} - \hat{s}_i + \hat{t}_j$ for $(i, j) \in E$; in particular, $\max_{(i,j)\in E}(\Delta s_i - \Delta t_j) \geq \varepsilon$ must hold. Thus, the $\ell^{\pm}_{\infty}$-distance from $(\hat{s}, \hat{t})$ to any point in $\mathrm{conv}(\mathrm{dom}\, g)$ is lower bounded by $\varepsilon$ as follows:

$$\|(\Delta s, \Delta t)\|^{\pm}_{\infty} \geq \max_{i \in L} \max\{0, +\Delta s_i\} + \max_{j \in R} \max\{0, -\Delta t_j\} \geq \max_{(i,j)\in E}(\Delta s_i - \Delta t_j) \geq \varepsilon.$$

This lower bound is attained by setting $\Delta s = \frac{\varepsilon}{2}\mathbf{1}$ and $\Delta t = -\frac{\varepsilon}{2}\mathbf{1}$, as in the lemma statement. $\qquad\square$

**Local optimization.** As in [17], local optimization reduces to the minimum vertex cover problem (the dual of the maximum cardinality matching problem). We below briefly describe this reduction. Given a current solution $p = (s, t) \in \mathrm{dom}\, g$, Step 3 asks to find an optimal direction $d = (\mathbf{1}_X, \mathbf{1}_Y)$ over $X \subseteq L$ and $Y \subseteq R$. Letting $S = X$ and $T = R \setminus Y$, we can formulate this problem as

$$\underset{S \subseteq L, T \subseteq R}{\text{minimize}} \ |S| + |T| + \text{const.} \quad \text{subject to} \ \ \mathbb{1}_{i \in S} + \mathbb{1}_{j \in T} \geq w_{ij} - s_i + t_j + 1 \quad \forall(i, j) \in E, \ (2)$$

where $\mathbb{1}_{\{\cdot\}} = 1\ (0)$ if the argument is true (false). The constraint for each $(i, j) \in E$ matters only when $(i, j)$ is *tight*, i.e., $s_i - t_j = w_{ij}$. Thus, letting $E^*$ be the set of tight edges, we can write (2) as

$$\underset{S \subseteq L, T \subseteq R}{\text{minimize}} \ |S| + |T| + \text{const.} \quad \text{subject to} \ \ i \in S \text{ or } j \in T \quad \forall(i, j) \in E^*,$$

which is the minimum vertex cover problem on $(V, E^*)$, the dual of maximum cardinality matching. If we solve it with the Hopcroft–Karp algorithm [26], we have $T_{\mathrm{loc}} = \mathrm{O}(m\sqrt{n})$.

By the Kőnig–Egerváry theorem [44, Theorem 16.2], there exists a vertex cover $(S, T)$ with $|S| + |T| < n/2$ if and only if $(V, E^*)$ has no perfect matching. Once a minimum vertex cover $(S, T)$ with $|S| + |T| < n/2$ is found, we update $(s, t)$ to $(s + \lambda\mathbf{1}_S, t + \lambda\mathbf{1}_{R\setminus T})$, where the step length $\lambda$ in Step 6 is given by $\min_{i \in L\setminus S, j \in R\setminus T}\{s_i - t_j - w_{ij}\}$. If we find a vertex cover $(S, T)$ such that $|S| + |T| = n/2$ on $(V, E^*)$, any maximum-cardinality matching on $(V, E^*)$ is a maximum-weight matching on $G$ by complementary slackness [44, Section 18.5b], thus solving the original problem.

## 3.2 Weighted matroid intersection

We next consider the weighted matroid intersection problem, a generalization of various problems such as bipartite matchings, packing spanning trees, and arborescences in a directed graph. Due to this broad coverage, the discussion here would serve as a role model for applying our DCA-based framework to various problems, even though the general result by itself may not immediately provide practical algorithms for specific problems.

A *matroid* $\mathbf{M}$ consists of a finite set $V$ and a non-empty set family $\mathcal{B} \subseteq 2^V$ satisfying the following exchange axiom: for any $B_1, B_2 \in \mathcal{B}$ and $i \in B_1 \setminus B_2$, there exists $j \in B_2 \setminus B_1$ such that $B_1 \setminus \{i\} \cup \{j\} \in \mathcal{B}$ and $B_2 \setminus \{j\} \cup \{i\} \in \mathcal{B}$. Elements in $\mathcal{B}$ are called *bases*, and an *independent set* is a subset of a base. The *rank function* $\rho : 2^V \to \mathbb{Z}$ of $\mathbf{M}$ is defined as $\rho(X) = \max_{B \in \mathcal{B}} |X \cap B|$. The *rank* of $\mathbf{M}$ is defined by $\rho(V)$, which coincides with the cardinality of any base $B \in \mathcal{B}$.

Let $\mathbf{M}_1 = (V, \mathcal{B}_1)$ and $\mathbf{M}_2 = (V, \mathcal{B}_2)$ be rank-$r$ matroids on an identical fixed ground set $V$ of size $n$, equipped with a weight vector $w \in \mathbb{Z}^V$. We assume that the matroids are given as independence oracles, which test whether an input set is independent or not in $\tau$ time. We also assume $\mathcal{B}_1 \cap \mathcal{B}_2 \neq \emptyset$ (which we can check by solving the maximum cardinality matroid intersection problem once). The weighted matroid intersection problem on $(\mathbf{M}_1, \mathbf{M}_2)$ asks to find $B \in \mathcal{B}_1 \cap \mathcal{B}_2$ that maximizes $w(B)$, where $v(X) = \sum_{i \in X} v_i$ for any $v \in \mathbb{Z}^V$ and $X \subseteq V$. Its dual structure can be captured via the weight-splitting theorem [22, Theorem 13.2.4] and the dual problem is written as

$$\underset{p \in \mathbb{Z}^V}{\text{minimize}} \quad g(p) = \max_{B \in \mathcal{B}_1} p(B) + \max_{B \in \mathcal{B}_2}(w - p)(B). \tag{3}$$

The objective function $g$ in (3) is known to be L-convex.[4] Since (3) is an unconstrained problem, nothing is needed for the projection. We below focus on the local optimization step (Step 3).

---

[4]For $k = 1, 2$, $g_k(p) = \max_{B \in \mathcal{B}_k} p(B)$ is an L-convex function obtained as the *discrete Legendre–Fenchel conjugate* of the indicator function $\delta_{\mathcal{B}_k}$ of $\mathcal{B}_k$ (regarded as the collection of $\mathbf{1}_B$ for $B \in \mathcal{B}_k$), which is *M-convex*. As $g_1$ and $g_2$ are L-convex, so is $g(p) = g_1(p) + g_2(w - p)$. See [38] for details.

**Local optimization.** We see that the local optimization reduces to the maximum cardinality matroid intersection problem, which asks to find a maximum-cardinality common independent set of two matroids. Letting $p \in \mathbb{Z}^V$ be a current feasible solution to (3) and $q = w - p$, the problem of finding an optimal direction $d = \mathbf{1}_X$ over $X \subseteq V$ is written as

$$\underset{X \subseteq V}{\text{minimize}} \quad \max_{B \in \mathcal{B}_1}(p + \mathbf{1}_X)(B) + \max_{B \in \mathcal{B}_2}(q - \mathbf{1}_X)(B). \tag{4}$$

For a matroid $\mathbf{M} = (V, \mathcal{B})$ and $v \in \mathbb{Z}^V$, let $\mathcal{B}^v = \operatorname{argmax}_{B \in \mathcal{B}} v(B)$ and $\mathbf{M}^v = (V, \mathcal{B}^v)$. Then, $\mathbf{M}^v$ forms a matroid [20], and its rank function is given as follows (see Appendix B.1 for the proof).

**Lemma 3.** *The rank function of $\mathbf{M}^v$ is given by $\rho^v(X) = \max_{B \in \mathcal{B}}(v + \mathbf{1}_X)(B) - \max_{B \in \mathcal{B}} v(B)$.*

From Lemma 3, by using rank functions $\rho_1^p$ and $\rho_2^q$ of $\mathbf{M}_1^p$ and $\mathbf{M}_2^q$, respectively, we can rewrite (4) as

$$\underset{X \subseteq V}{\text{minimize}} \quad \rho_1^p(X) + \rho_2^q(V \setminus X) + \text{const.}, \tag{5}$$

where const. $= \max_{B \in \mathcal{B}_1} p(B) + \max_{B \in \mathcal{B}_2} q(B) - r$. The problem (5) (without the constant term) is the dual of the maximum cardinality matroid intersection problem on $(\mathbf{M}_1^p, \mathbf{M}_2^q)$ (Edmonds' matroid intersection theorem [19]), thus completing the reduction. The standard augmenting-path algorithm by Cunningham [15] makes $\mathrm{O}(nr^{1.5})$ calls to independence oracles of $\mathbf{M}_1^p$ and $\mathbf{M}_2^q$, and every independence testing on $\mathbf{M}_1^p$ and $\mathbf{M}_2^q$ queried by the algorithm can be implemented with a single call to independence oracles of $\mathbf{M}_1$ and $\mathbf{M}_2$, respectively, which takes $\tau$ time (see Appendix B.2).

**Lemma 4.** *There is an algorithm that solves the maximum cardinality matroid intersection problem on $(\mathbf{M}_1^p, \mathbf{M}_2^q)$ by making $\mathrm{O}(nr^{1.5})$ calls to independence oracles of $\mathbf{M}_1$ and $\mathbf{M}_2$.*

Once a maximum common independent set $I$ of $(\mathbf{M}_1^p, \mathbf{M}_2^q)$ is found, we can obtain optimal solution $X \subseteq V$ to (5) by traversing the auxiliary graph constructed in the augmenting-path algorithm (see [44, Theorem 41.3]). Thus, the total time complexity of local optimization is $T_{\text{loc}} = \mathrm{O}(\tau n r^{1.5})$. If $|I| < r$, we update $p$ to $p + \lambda \mathbf{1}_X$, where the step length $\lambda$ is determined by binary search [46] (or let $\lambda = 1$). If $|I| = r$, current $p$ is optimal to (3) and $I$ is a maximum weight common base of $(\mathbf{M}_1, \mathbf{M}_2)$ [21].

### 3.3 Discrete energy minimization

We consider discrete energy minimization problems on a fixed vertex set $V$. Let $G = (V, E)$ be an undirected graph with $|V| = n$ and $|E| \le m$. Given univariate convex functions $\phi_i : \mathbb{Z} \to \mathbb{R} \cup \{+\infty\}$ and $\psi_{ij} : \mathbb{Z} \to \mathbb{R} \cup \{+\infty\}$ for $i, j \in V$ with $i \ne j$, the energy minimization problem is written as

$$\underset{p \in \mathbb{Z}^V}{\text{minimize}} \quad g(p) = \sum_{i \in V} \phi_i(p_i) + \sum_{(i,j) \in E} \psi_{ij}(p_j - p_i). \tag{6}$$

This problem appears in many computer-vision (CV) applications [8, 25, 28] and belongs to $\mathrm{L}^\natural$-convex minimization [32]. In CV settings, $\phi_i$ measures how well label $p_i$ fits pixel $i$, and $\psi_{ij}$ quantifies smoothness of labels of adjacent pixels $i$ and $j$.

**Projection.** In CV applications, we usually have a box constraint representing the range of pixel values. Also, we may have an acceptable range of non-smoothness of adjacent pixels. To deal with such constraints, we suppose $\phi_i$ and $\psi_{ij}$ to return $+\infty$ if input variables are out of the ranges. The resulting feasible region can be represented as in Proposition 1. If we only have box constraints, we can easily obtain an $\ell_\infty^\pm$-projection of $\hat{p} \in \mathbb{R}^V$ by computing $\max\{\alpha_i, \min\{\hat{p}_i, \beta_i\}\}$ for each $i \in V$, which takes $T_{\text{prj}} = \mathrm{O}(n)$ time. When imposing the smoothness constraints, we need to compute an $\ell_\infty^\pm$-projection onto a general $\mathrm{L}^\natural$-convex set. This is reduced to the shortest path problem, and we can compute the projection in $T_{\text{prj}} = \mathrm{O}(mn)$ time with the Bellman–Ford algorithm (see Appendix D).

**Local optimization.** Since the steepest descent method for problem (6) is already studied in [32], we here briefly describe key points. Let $p \in \operatorname{dom} g$ be a current solution and consider finding a steepest direction $d \in \mathcal{N}_\pm = \{0, +1\}^V \cup \{0, -1\}^V$. We focus on exploring $\{0, +1\}^V$; the case of $\{0, -1\}^V$ is analogous. Letting $\phi_i^{(p)}(d_i) = \phi_i(p_i + d_i)$ for $i \in V$ and $\psi_{ij}^{(p)}(d_i, d_j) = \psi_{ij}(p_j + d_j - p_i - d_i)$ for $(i, j) \in E$, the local optimization problem on $\{0, +1\}^V$ is written as

$$\underset{d \in \{0, +1\}^V}{\text{minimize}} \quad g^{(p)}(d) = \sum_{i \in V} \phi_i^{(p)}(d_i) + \sum_{(i,j) \in E} \psi_{ij}^{(p)}(d_i, d_j).$$

Since convexity of $\psi_{ij}$ implies $\psi_{ij}^{(p)}(1,0) + \psi_{ij}^{(p)}(0,1) \geq \psi_{ij}^{(p)}(1,1) + \psi_{ij}^{(p)}(0,0)$, $g^{(p)}(d)$ is a submodular function that can be written as a sum of pseudo-boolean functions with at most two variables. Minimization of such a submodular function is reduced to a min-cut problem on a graph with $n + 2$ vertices and $3m + n$ edges [29, 6]. If we solve the min-cut problem with the Dinic algorithm [16], it takes $T_{\text{loc}} = \mathrm{O}(mn^2)$ time. We can also use empirically fast min-cut algorithms for CV settings [7]. Once a direction $d$ is found, the step length $\lambda$ can be determined by binary search [46] (or let $\lambda = 1$).

### 3.4    General $\mathrm{L}^\natural$-convex function minimization

We consider general $\mathrm{L}^\natural$-convex function minimization $\min_{p \in \mathbb{Z}^V} g(p)$, assuming $\mathrm{dom}\, g$ to be represented as in Proposition 1. Given a prediction $\hat{p} \in \mathbb{R}^V$, we can compute an $\ell_\infty^\pm$-projection $\mathcal{P}(\hat{p})$ onto $\mathrm{conv}(\mathrm{dom}\, g)$ with the Bellman–Ford algorithm in $T_{\text{prj}} = \mathrm{O}(n^3)$ time (since $m \leq n^2$), as mentioned in Section 3.3. As for the local optimization, given a current solution $p \in \mathrm{dom}\, g$, we can find a steepest direction by solving $\min_{X \subseteq V} f(X)$, where $f(X) = g(p + \mathbf{1}_X)$ or $g(p - \mathbf{1}_X)$ is a submodular function, as mentioned in Section 2.2. An empirically fast algorithm for submodular function minimization is the Fujishige–Wolfe algorithm [23], which solves the local optimization problem in $T_{\text{loc}} = \mathrm{O}((n^4\mathrm{EO} + n^5)F^2)$ time [11, 33], where EO is the time of evaluating $f(X)$ for any $X \subseteq V$ and $F = \max_{i \in V} \max\{|f(\{i\})|, |f(V) - f(V \setminus \{i\})|\}$ (and a theoretically faster strongly polynomial-time algorithm runs in $\mathrm{O}(n^3 \log^2 n \cdot \mathrm{EO} + n^4 \log^{\mathrm{O}(1)} n)$ time [35]). In total, we can minimize a general $\mathrm{L}^\natural$-convex function in $\mathrm{O}((n^4\mathrm{EO} + n^5)F^2 \cdot \|p^*(g) - \hat{p}\|_\infty)$ time.

## 4    Learning predictions

We detail how to learn predictions $\hat{p} \in \mathbb{R}^V$. We consider online and batch learning settings where instances of form $\min_{p \in \mathbb{Z}^V} g_t(p)$ for $t = 1, \ldots, T$ are given adversarially and randomly, respectively. We make the following assumption, which is common with the previous studies [17, 31].

**Assumption 2.** *Functions $g_1, \ldots, g_T$ are defined on the same ground set $V$ and satisfy Assumption 1.*

Furthermore, as in Theorem 2, we assume that a benchmark (optimal) prediction $\hat{p}^*$ is included in $[-C, +C]^V$ for some $C > 0$, which is also common with [17, 31]. As for bipartite matching, there always exists an optimal dual solution included in $[-C, +C]^V$ if $C \geq n\|w\|_\infty$ [30, Lemma 4.5],[5] and the same condition holds for matroid intersection with $C \geq r\|w\|_\infty$ (see Appendix B.3). This condition is also natural in discrete energy minimization since the range of pixel values is bounded. Also, we can choose any suboptimal solution in $[-C, +C]^V$ as a benchmark prediction at the cost of increasing the bounds in Theorem 2.

We use the online-learning framework by Khodak et al. [31], and thus the learning method itself is not novel. Nevertheless, the improvement in the bounds is considerable. As with [31], the regret bound is obtained from that of the online gradient descent method (OGD) [45], and the sample complexity bound follows from online-to-batch conversion [10]. The key to the improvement is that we apply OGD to 1-Lipschitz convex loss functions $L_t(p) = \|p^*(g_t) - p\|_\infty$,[6] while Khodak et al. [31] applied OGD to $L_t(p) = \|p^*(g_t) - p\|_1$, which is $\sqrt{n}$-Lipschitz, following the setting of [17].

**Theorem 2.** *Let $V$ be a finite set of size $n$ and $C > 0$. For any sequence of functions $g_1, \ldots, g_T$ that map $\mathbb{Z}^V$ to $\mathbb{R} \cup \{+\infty\}$, there is an online learning algorithm that returns $\hat{p}_1, \ldots, \hat{p}_T \in \mathbb{R}^V$ with the following regret bound for any $\hat{p}^* \in [-C, +C]^V$:*

$$\sum_{t=1}^{T} \|p^*(g_t) - \hat{p}_t\|_\infty \leq \sum_{t=1}^{T} \|p^*(g_t) - \hat{p}^*\|_\infty + C\sqrt{2nT}.$$

*Moreover, for any distribution $\mathcal{D}$ over functions $g : \mathbb{Z}^V \to \mathbb{R} \cup \{+\infty\}$, $\delta \in (0, 1]$, and $\varepsilon > 0$, if $T = \Omega\left(\left(\frac{C}{\varepsilon}\right)^2 \left(n + \log \frac{1}{\delta}\right)\right)$ i.i.d. samples $g_1, \ldots, g_T$ from $\mathcal{D}$ are given, we can compute $\hat{p} \in \mathbb{R}^V$ that satisfies the following condition with a probability of at least $1 - \delta$ for all $\hat{p}^* \in [-C, +C]^V$:*

$$\mathbb{E}_{g \sim \mathcal{D}} \|p^*(g) - \hat{p}\|_\infty \leq \mathbb{E}_{g \sim \mathcal{D}} \|p^*(g) - \hat{p}^*\|_\infty + \varepsilon.$$

---

[5]In [17], it is stated that $C \geq \|w\|_\infty$ is sufficient, but this is not true as shown in Appendix C.

[6]We can also use $L_t(p) = \|p^*(g_t) - p\|_\infty^\pm$, which is convex and $\sqrt{2}$-Lipschitz; this may yield shaper bounds on the number of iterations, as in Proposition 2. We here present the $\ell_\infty$-loss version for ease of exposition.

*Proof.* We regard $L_t(p) = \|p^*(g_t) - p\|_\infty$ for $t = 1, \ldots, T$ as loss functions of $p \in \mathbb{R}^V$ and use the online gradient descent method (OGD). Note that $L_t(p)$ is convex since

$$L_t\left(\frac{p+q}{2}\right) = \left\|p^*(g_t) - \frac{p+q}{2}\right\|_\infty \leq \left\|\frac{p^*(g_t)}{2} - \frac{p}{2}\right\|_\infty + \left\|\frac{p^*(g_t)}{2} - \frac{q}{2}\right\|_\infty = \frac{L_t(p) + L_t(q)}{2}$$

for any $p, q \in \mathbb{R}^V$ due to the triangle inequality. Furthermore, $L_t(p)$ is 1-Lipschitz since

$$L_t(p) - L_t(q) = \|p^*(g_t) - p\|_\infty - \|p^*(g_t) - q\|_\infty \leq \|p - q\|_\infty \leq \|p - q\|_2$$

for any $p, q \in \mathbb{R}^V$ due to the triangle inequality and $\|x\|_\infty \leq \|x\|_2$ for any $x \in \mathbb{R}^V$. Since $L_t$ is a 1-Lipschitz convex loss function and the $\ell_2$-norm of any vector in $[-C, +C]^V$ is at most $C\sqrt{n}$, the regret of OGD is at most $C\sqrt{2nT}$ (see [45, Corollary 2.7]), thus obtaining the first claim. The second claim is obtained by using online-to-batch conversion. Specifically, since the loss function value is at most $2C$, if we feed sampled $g_1, \ldots, g_T$ to OGD and let $\hat{p} = \frac{1}{T}\sum_{t=1}^T \hat{p}_t$, then [10, Proposition 1] implies that the following inequality holds with a probability of at least $1 - \delta$:

$$\mathbb{E}_{g \sim \mathcal{D}}\|p^*(g) - \hat{p}\|_\infty \leq \min_{p \in [-C,+C]^V} \mathbb{E}_{g \sim \mathcal{D}}\|p^*(g) - p\|_\infty + C\sqrt{\frac{2n}{T}} + 2C\sqrt{\frac{2}{T}\log\frac{1}{\delta}}.$$

Therefore, setting $T = 32\left(\frac{C}{\varepsilon}\right)^2\left(n + \log\frac{1}{\delta}\right)$ is sufficient for bounding the sum of the last two terms in the right-hand side by $\varepsilon$ from above. $\quad\square$

# 5 Whether to learn primal or dual solutions

We discuss whether to learn primal or dual solutions for successfully warm-starting algorithms with predictions from the DCA perspective (we refer the reader to [38] for more information). We expect that the discussion here is also useful in the context of augmenting online algorithms with predictions.

We consider a minimization problem of form $\min_{p \in \mathbb{Z}^V} g(p)$, where $g : \mathbb{Z}^V \to \mathbb{R} \cup \{+\infty\}$ is a general objective function. As we have seen above, iterative algorithms look for an optimal solution by alternately exploring a neighborhood to find a steepest direction and proceeding along the direction. Intuitively, such an iterative algorithm can benefit from a prediction that is close to an optimum if the feasible region, $\mathrm{dom}\, g$, is *path connected* with respect to the neighborhood; conversely, if the feasible region is not connected, good predictions are not always helpful. We below elaborate more on this idea for the case of L-convex minimization (the $\mathrm{L}^\natural$-convex case is analogous).

First, we need to define a neighborhood $\mathcal{N}$ so that we can efficiently solve local optimization problems on $\mathcal{N}$. In the case of L-convex minimization, the local optimization on $\mathcal{N} = \mathcal{N}_+ = \{0, +1\}^V$ reduces to submodular function minimization in general, which we can solve in polynomial time (and more efficient methods are available for many specific problems). Once a neighborhood $\mathcal{N}$ is defined, the next important requirement is the *path connectivity* of the feasible region $\mathrm{dom}\, g$ with respect to $\mathcal{N}$, i.e., given any $p, q \in \mathrm{dom}\, g$, we can move from $p$ to $q$ by iteratively finding an appropriate direction $d \in \mathcal{N}$ and proceeding along $d$. L-convex sets are path connected with respect to $\mathcal{N}_+$, and this property is necessary for ensuring that the steepest descent method converges to an optimum.

Under the above conditions, a prediction close to an optimum is expected to be beneficial since it provides a short path to an optimum. In other words, if feasible regions are not path connected, predictions close to an optimum do not always improve the performance of iterative algorithms. This observation gives a guideline for judging whether to learn primal or dual solutions: we should choose the one such that a prediction can be converted into a solution in a path-connected feasible region.

In DCA, there are other convexity notions than L-convexity: $M$-, $L_2$-, and $M_2$-*convexity* (and their $\mathrm{L}^\natural$ counterparts). In these classes, L-convex sets are path connected with respect to $\mathcal{N}_+$, and so are M-convex sets with respect to $\mathcal{N} = \left\{\mathbf{1}_{\{i\}} - \mathbf{1}_{\{j\}} \mid i, j \in V\right\}$. By contrast, no appropriate neighborhoods are known that make $L_2$- and $M_2$-convex sets path connected. In the cases of bipartite matching and matroid intersection, the primal problems belong to $M_2$-concave maximization, while their dual problems are L-convex minimization. Thus, learning dual solutions for bipartite matching and matroid intersection is reasonable. On the other hand, the primal problem of discrete energy minimization is L-convex, and thus we do not need to consider its dual.

## 6    Conclusion

We have developed a framework for warm-starting the steepest descent method for L/L$^\natural$-convex function minimization with predictions, thus bridging between discrete convex analysis and algorithms with predictions. We have demonstrated its effectiveness for weighted bipartite matching, weighted matroid intersection, and discrete energy minimization. We have also presented regret and sample complexity bounds for learning of predictions and discussed whether to learn primal or dual solutions. As for the practical aspect, experiments in Appendix F show that our DCA-based approach performs comparably to (or slightly better than) the method of [17].

A limitation of our work is that it does not yield prediction-independent worst-case bounds in general. This, however, is often not a serious matter since we can run standard algorithms with worst-case guarantees in parallel and terminate both once either one returns a solution. This point should be contrasted with the situation of augmenting online algorithms with predictions, where every decision made over time is irrevocable and thus attaining good prediction-dependent and worst-case guarantees simultaneously by a single algorithm is crucial. In Appendix E, we further discuss worst-cae guarantees of our DCA-based framework. In addition, we cannot deal with problems without L/L$^\natural$-convexity, and extending our framework to M/M$^\natural$-convex function minimization, which also enjoys the path connectivity as mentioned in Section 5, will be an interesting future direction.

## Acknowledgements

The authors thank Mikhail Khodak for sharing the latest results of [31] and anonymous reviewers for their valuable suggestions. This work was supported by JST ERATO Grant Number JPMJER1903 and JSPS KAKENHI Grant Number JP22K17853.

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
