*Proof.* We focus on the case where $S$ is L$^\natural$-convex; the following proof subsumes the case where $S$ is L-convex. From Proposition 1, $S$ can be written as

$$S = \left\{ p \in \mathbb{Z}^V \;\middle|\; \alpha_i \leq p_i \leq \beta_i, \;\; p_j - p_i \leq \gamma_{ij} \;\text{ for } i,j \in V; i \neq j \right\}$$

with some $\alpha_i \in \mathbb{Z} \cup \{-\infty\}$, $\beta_i \in \mathbb{Z} \cup \{+\infty\}$, and $\gamma_{ij} \in \mathbb{Z} \cup \{+\infty\}$. Also, $\operatorname{conv}(S)$ has the same representation except for the replacement of $\mathbb{Z}^V$ with $\mathbb{R}^V$. We prove that for any $p \in \mathbb{R}^V$ satisfying the inequality constraints, $\lfloor p \rceil \in \mathbb{Z}^V$ also satisfies the constraints, which implies the lemma statement.

Since $\alpha_i \in \mathbb{Z} \cup \{-\infty\}$ and $\beta_i \in \mathbb{Z} \cup \{+\infty\}$, $\alpha_i \leq p_i \leq \beta_i$ implies $\alpha_i \leq \lfloor p_i \rceil \leq \beta_i$. We below discuss the remaining constraints of form $p_j - p_i \leq \gamma_{ij}$. Since $p_j - p_i \leq \gamma_{ij}$ and $\gamma_{ij} \in \mathbb{Z} \cup \{+\infty\}$ imply $\lceil p_j - p_i \rceil \leq \gamma_{ij}$, it suffices to prove $\lfloor p_j \rceil - \lfloor p_i \rceil \leq \lceil p_j - p_i \rceil$. Let $p_i = a + r$ and $p_j = b + s$, where $a, b \in \mathbb{Z}$ and $r, s \in [0, 1)$. If $s > r$, since $\lfloor p_i \rceil \geq a$ and $\lfloor p_j \rceil \leq b + 1$, we have

$$\lfloor p_j \rceil - \lfloor p_i \rceil \leq b + 1 - a = \lceil b - a + s - r \rceil = \lceil p_j - p_i \rceil.$$

If $s \leq r$, we have

$$\lfloor p_j \rceil - \lfloor p_i \rceil = \lfloor b + s \rceil - \lfloor a + r \rceil \leq \lfloor b + r \rceil - \lfloor a + r \rceil = b - a = \lceil b - a + s - r \rceil = \lceil p_j - p_i \rceil.$$

In any case, we have $\lfloor p_j \rceil - \lfloor p_i \rceil \leq \lceil p_j - p_i \rceil \leq \gamma_{ij}$. Hence $p \in \operatorname{conv}(S)$ implies $\lfloor p \rceil \in S$. $\qquad\square$

## B   Details of matroid intersection

We describe details of the weighted matroid intersection algorithm discussed in Section 3.2 and present proofs of Lemmas 3 and 4. We also give a bound on dual optimal solutions to the weighted matroid intersection problem. Readers interested in matroid theory are referred to [40].

### B.1   Proof of Lemma 3

Let $\mathbf{M} = (V, \mathcal{B})$ be a matroid and $v \in \mathbb{Z}^V$ be a weight vector. Recall that $\mathbf{M}^v = (V, \mathcal{B}^v)$ denotes the matroid with $\mathcal{B}^v = \operatorname{argmax}_{B \in \mathcal{B}} v(B)$. A maximum-weight base $B \in \mathcal{B}^v$ can be obtained by the following greedy algorithm [20]. First, sort elements of $V$ in a non-increasing order of $v$ as $i_1, \ldots, i_{|V|}$, i.e., $v_{i_1} \geq \cdots \geq v_{i_{|V|}}$. Starting with $B = \emptyset$, from $k = 1$ to $|V|$, add $i_k$ to $B$ if $B \cup \{i_k\}$ is independent. The resulting $B$ is a base of $\mathbf{M}$ that maximizes $v(B)$. The correctness of this greedy algorithm justifies Lemma 3 as follows.

**Lemma 3.** *The rank function of $\mathbf{M}^v$ is given by $\rho^v(X) = \max_{B \in \mathcal{B}} (v + \mathbf{1}_X)(B) - \max_{B \in \mathcal{B}} v(B)$.*

*Proof.* By the definitions of the rank function and $\mathcal{B}^v$, we have

$$\rho^v(X) = \max_{B \in \mathcal{B}^v} |X \cap B| = \max_{B \in \mathcal{B}^v} \mathbf{1}_X(B) = \max_{B \in \mathcal{B}^v} (v + \mathbf{1}_X)(B) - \max_{B \in \mathcal{B}} v(B).$$

Therefore, our goal is to show $\max_{B \in \mathcal{B}} (v + \mathbf{1}_X)(B) = \max_{B \in \mathcal{B}^v} (v + \mathbf{1}_X)(B)$. Consider an ordering of elements in $V$ that is non-increasing with respect to $v + \frac{1}{2}\mathbf{1}_X$. Since $v$ is an integer vector, this order is non-increasing with respect to both $v$ and $v + \mathbf{1}_X$, meaning that there exists $B \in \mathcal{B}$ that maximizes $v$ and $v + \mathbf{1}_X$ simultaneously, hence $\max_{B \in \mathcal{B}} (v + \mathbf{1}_X)(B) = \max_{B \in \mathcal{B}^v} (v + \mathbf{1}_X)(B)$. $\qquad\square$

### B.2   Proof of Lemma 4

We give preliminaries on matroid theory (see [40] for details). Let $\mathbf{M} = (V, \mathcal{B})$ be a matroid and $X$ be a subset of $V$. The *restriction* of $\mathbf{M}$ to $X$, denoted by $\mathbf{M} \mid X$, is the matroid on $X$ whose independent sets are subsets of $X$ that are independent in $\mathbf{M}$. The *contraction* of $\mathbf{M}$ with respect to $X$ is the matroid $\mathbf{M} / X = (V \setminus X, \mathcal{B} / X)$ with $\mathcal{B} / X = \{ B \subseteq V \setminus X \mid B \cup B' \in \mathcal{B} \}$, where $B'$ is any base of $\mathbf{M} \mid X$ (indeed, $\mathbf{M} / X$ does not depend on the choice of $B'$). The *direct sum* of two matroids

$\mathbf{M}_1 = (V_1, \mathcal{B}_1)$ and $\mathbf{M}_2 = (V_2, \mathcal{B}_2)$ with $V_1 \cap V_2 = \emptyset$ is the matroid $\mathbf{M}_1 \oplus \mathbf{M}_2 = (V_1 \cup V_2, \mathcal{B}_1 \oplus \mathcal{B}_2)$ with $\mathcal{B}_1 \oplus \mathcal{B}_2 = \{ B_1 \cup B_2 \mid B_1 \in \mathcal{B}_1, B_2 \in \mathcal{B}_2 \}$.

Define $V^v_{\geq}(t) = \{ i \in V \mid v_i \geq t \}$ for any $t \in \mathbb{Z}$. The correctness of the greedy algorithm for $v \in \mathbb{Z}^V$ also gives the following well-known decomposition theorem of $\mathbf{M}^v$ (see, e.g., [27, Lemma 1]).

**Lemma 5.** *It holds* $\mathbf{M}^v = \bigoplus_{t=-\infty}^{\infty} \mathbf{M}^v(t)$, *where* $\mathbf{M}^v(t) = \left( \mathbf{M} \mid V^v_{\geq}(t) \right) / V^v_{\geq}(t+1)$.

We are ready to prove Lemma 4.

**Lemma 4.** *There is an algorithm that solves the maximum cardinality matroid intersection problem on* $(\mathbf{M}^p_1, \mathbf{M}^q_2)$ *by making* $\mathrm{O}(nr^{1.5})$ *calls to independence oracles of* $\mathbf{M}_1$ *and* $\mathbf{M}_2$.

*Proof.* Cunningham's algorithm [15] asks the independence of sets in the form $I \cup \{j\}$ or $I \setminus \{i\} \cup \{j\}$, where $I$ is a common independent set of $(\mathbf{M}^p_1, \mathbf{M}^q_2)$, $i \in I$, and $j \in V \setminus I$. Thus, it suffices to show the following: for a matroid $\mathbf{M} = (V, \mathcal{B})$, a weight vector $v \in \mathbb{Z}^V$, an independent set $I$ of $\mathbf{M}^v$, $i \in I$, and $j \in V \setminus I$, one can check whether $I \cup \{j\}$ and $I \setminus \{i\} \cup \{j\}$ are independent or not in $\mathbf{M}^v$ by a single call to the independence oracle of $\mathbf{M}$.

Let $V^v(t) = \{ i \in V \mid v_i = t \}$ for $t \in \mathbb{Z}$. By Lemma 5, a set $X \subseteq V$ is independent in $\mathbf{M}^v$ if and only if $X \cap V^v(t)$ is independent in $\mathbf{M}^v(t)$ for every $t \in \mathbb{Z}$. This and the independence of $I$ in $\mathbf{M}^v$ imply that $I \cup \{j\}$ and $I \setminus \{i\} \cup \{j\}$ are independent in $\mathbf{M}^v$ if and only if $(I \cup \{j\}) \cap V^v(v_j)$ and $(I \setminus \{i\} \cup \{j\}) \cap V^v(v_j)$, respectively, are independent in $\mathbf{M}^v(v_j)$. The independence of a set $X \subseteq V^v(t)$ in $\mathbf{M}^v(t)$ is equivalent to that of $X \cup (B \cap V^v_{\geq}(t+1))$ in $\mathbf{M}$, where $B$ is an arbitrary base of $\mathbf{M}^v$. Therefore, we can reduce the independence testing of $I \cup \{j\}$ and $I \setminus \{i\} \cup \{j\}$ in $\mathbf{M}^v$ to that of $(I \cup \{j\}) \cup (B \cap V^v_{\geq}(v_j + 1))$ and $(I \setminus \{i\} \cup \{j\}) \cup (B \cap V^v_{\geq}(v_j + 1))$, respectively, in $\mathbf{M}$. We can obtain $B \in \mathcal{B}^v$ by running the greedy algorithm once in advance, which makes $\mathrm{O}(n)$ calls to the independence oracle of $\mathbf{M}$. Thus, we obtain the desired oracle complexity. $\qquad \square$

### B.3 Bound on dual optimal solutions

Let $\mathbf{M}_1 = (V, \mathcal{B}_1)$ and $\mathbf{M}_2 = (V, \mathcal{B}_2)$ be matroids and $w \in \mathbb{Z}^V$ be a weight vector. Fixing any maximum-weight common base $B \in \mathcal{B}_1 \cap \mathcal{B}_2$, let $D = (V, A)$ be a directed bipartite graph with bipartition $V = B \cup (V \setminus B)$ and arc set $A = A_1 \cup A_2$ given by

$$
\begin{aligned}
A_1 &= \{ (i,j) \mid i \in B, j \in V \setminus B, B \setminus \{i\} \cup \{j\} \in \mathcal{B}_1 \}, \\
A_2 &= \{ (j,i) \mid i \in B, j \in V \setminus B, B \setminus \{i\} \cup \{j\} \in \mathcal{B}_2 \}.
\end{aligned}
\tag{7}
$$

We define an arc-length vector $\gamma \in \mathbb{Z}^A$ by $\gamma_{ij} = -w_j$ for $(i,j) \in A_1$ and $\gamma_{ji} = w_i$ for $(j,i) \in A_2$. Then, $D$ has no negative cycles [44, Theorem 41.5] and thus has a *potential* $p \in \mathbb{Z}^V$, which is a vector satisfying $p_j - p_i \leq \gamma_{ij}$ for $(i,j) \in A$. Indeed, the potentials coincide with the dual optimal solutions, as in the following lemma.

**Lemma 6** ([44, Theorem 41.9]). *A vector* $p \in \mathbb{Z}^V$ *is optimal to* (3) *if and only if* $p$ *is a potential of* $D$ *with respect to the arc-length vector* $\gamma \in \mathbb{Z}^A$.

This lemma gives a bound on an optimal dual solution to the weighted matroid intersection problem.

**Proposition 3.** *There exists a dual optimal solution* $p^* \in \mathbb{Z}^V$ *such that* $\|p^*\|_\infty \leq rW$, *where* $r$ *is the rank of* $\mathbf{M}_1$ *and* $\mathbf{M}_2$ *and* $W = \|w\|_\infty$.

*Proof.* Let $p \in \mathbb{Z}^V$ be the vector whose $i$th component $p_i$ is the minimum length of any directed paths on $D$ that ends at $i$ (starting from wherever). Then, $p$ is a potential of $D$ [44, Theorem 8.2] and thus is a dual optimal solution by Lemma 6. The number of arcs in a simple path on $D$ is at most $2r$, and each arc has a length at least $-W$, hence $-2rW \leq p_i \leq 0$. Let $p^* = p + rW\mathbf{1}$, which is also a dual optimal solution and satisfies $-rW \leq p^*_i \leq rW$ for $i \in V$. Hence, we have $\|p^*\|_\infty \leq rW$. $\qquad \square$

The following example shows that the bound given in Proposition 3 is tight.

**Example 1.** Let $n \in \mathbb{N}$ be odd. Define $\mathbf{M}_1 = (V, \mathcal{B}_1)$ and $\mathbf{M}_2 = (V, \mathcal{B}_2)$ by $V = \{1, \dots, n\}$ and

$$
\begin{aligned}
\mathcal{B}_1 &= \{ B \subseteq V \mid 1 \notin B \text{ and } |B \cap \{i, i+1\}| = 1 \text{ for even } i \in V \}, \\
\mathcal{B}_2 &= \{ B \subseteq V \mid n \notin B \text{ and } |B \cap \{i, i+1\}| = 1 \text{ for odd } i \in V \}.
\end{aligned}
$$

Then, $\mathbf{M}_1$ and $\mathbf{M}_2$ are matroids (*partition matroids*) of rank $r = (n-1)/2$ having a unique common base $B^* = \{2, 4, \ldots, n-3, n-1\}$. The sets $A_1$ and $A_2$ defined in (7) with respect to $B = B^*$ are $A_1 = \{(2,3), (4,5), \ldots, (n-1,n)\}$ and $A_2 = \{(1,2), (3,4), \ldots, (n-2, n-1)\}$, respectively, meaning that $D = (V, A)$ with $A = A_1 \cup A_2$ is the directed path graph from 1 to $n$. Let $W \in \mathbb{N}$ and $w \in \mathbb{Z}^V$ be a weight vector defined by $w_i = (-1)^{i+1} W$ for $i \in V$. Then, the corresponding arc length $\gamma \in \mathbb{Z}^A$ is $\gamma_{i,i+1} = -W$ for every arc $(i, i+1) \in A$.

Let $p^* \in \mathbb{Z}^V$ be a dual optimal solution. By Lemma 6, $p^*$ is a potential of $D$ with respect to $\gamma$. Thus, we have $p^*_{i+1} - p^*_i \leq -W$ for $i = 1, \ldots, n-1$, hence $p^*_i \leq p^*_1 - (i-1)W$ for $i \in V$. The $\ell_\infty$-norm of $p^*$ is minimized when $p^*_i = ((n+1)/2 - i)W$ for $i \in V$ with the minimum value of $(n-1)W/2 = rW$, thus implying the tightness of the bound in Proposition 3.

## C Counterexample to the bound on dual solutions for bipartite matching

In the proof of [17, Lemma 22], the following claim is used: if all edge costs $c_e$ of a bipartite graph $(V, E)$ are non-negative and at most $C$, there is an optimal dual solution $y^* \in \mathbb{R}^V$ to the *minimum cost perfect bipartite matching problem* such that $\|y^*\|_\infty \leq C$, where the dual problem is given as

$$\underset{y \in \mathbb{R}^V}{\text{maximize}} \quad \sum_{i \in V} y_i \quad \text{subject to} \quad y_i + y_j \leq c_e \quad (e = \{i, j\} \in E).$$

We give a counterexample to this claim.

**Example 2.** Let $n \in \mathbb{N}$ be even and $P_n = (V, E)$ be the path graph with vertices $V = \{1, \ldots, n\}$ and edges $E = \{\{i, i+1\} \mid i = 1, \ldots, n-1\}$. Then, $P_n$ is a bipartite graph with bipartition $\{L, R\}$ of $V$, where $L$ and $R$ consist of the odd and even numbers in $V$, respectively. Set an edge cost $c_e$ of $e = \{i, i+1\} \in E$ to $C > 0$ if $i \in L$ and to 0 if $i \in R \setminus \{n\}$. Since $M^* = \{\{i, i+1\} \mid i \in L\}$ is a unique perfect matching, any optimal dual solution $y^*$ must satisfy the tightness condition for all edges in $M^*$, i.e., $y^*_i + y^*_{i+1} = C$ for $i \in L$. In addition, the feasibility of $y^*$ implies $y^*_i + y^*_{i+1} \leq 0$ for $i \in R \setminus \{n\}$. Thus, we have $y^*_i \geq y^*_{i+2} + C$ for $i \in L \setminus \{n-1\}$, hence $y^*_1 \geq y^*_{n-1} + (n/2 - 1)C$. For $n \geq 8$, $y^*_{n-1} \in [-C, +C]$ implies $y^*_1 \geq 2C > C$, contradicting the claim.

## D Projection onto general $\mathrm{L}^\natural$-convex sets

Let $S \subseteq \mathbb{Z}^V$ be a non-empty $\mathrm{L}^\natural$-convex set. We assume that $S$ is represented as in Proposition 1, i.e.,

$$S = \left\{ p \in \mathbb{Z}^V \mid \alpha_i \leq p_i \leq \beta_i, \ p_j - p_i \leq \gamma_{ij} \text{ for } i, j \in V; i \neq j \right\}$$

where $\alpha_i \in \mathbb{Z} \cup \{-\infty\}$, $\beta_i \in \mathbb{Z} \cup \{+\infty\}$, and $\gamma_{ij} \in \mathbb{Z} \cup \{+\infty\}$. We define an edge set $E = \{(i, j) \mid i, j \in V; i \neq j, \gamma_{ij} < +\infty\}$ and let $m = |E|$ (note that constraints with $\gamma_{ij} = +\infty$ can be ignored). Given any $\hat{p} \in \mathbb{R}^V$, we show that an $\ell_\infty^\pm$-projection $\mathcal{P}(\hat{p}) \in \operatorname{argmin}_{p \in \operatorname{conv}(S)} \|p - \hat{p}\|_\infty^\pm$ can be computed in $\mathrm{O}(mn)$ time, where $n = |V|$.

Using variables $q = p - \hat{p} \in \mathbb{R}^V$, we rewrite the problem of computing $\mathcal{P}(\hat{p})$ as

$$\begin{aligned}
\text{minimize} \quad & \|q\|_\infty^\pm = \max_{i \in V} \max\{0, +q_i\} + \max_{i \in V} \max\{0, -q_i\} \\
\text{subject to} \quad & \alpha_i - \hat{p}_i \leq q_i \leq \beta_i - \hat{p}_i \quad \forall i \in V \\
& q_j - q_i \leq \gamma_{ij} - \hat{p}_j + \hat{p}_i \quad \forall i, j \in V; i \neq j.
\end{aligned}$$

For convenience, let $\hat{\gamma}_{i0} = -\alpha_i + \hat{p}_i$, $\hat{\gamma}_{0i} = \beta_i - \hat{p}_i$, and $\hat{\gamma}_{ij} = \gamma_{ij} - \hat{p}_j + \hat{p}_i$ for $i, j \in V$ such that $i \neq j$, $V_0 = \{0\} \cup V$, and $E_0 = E \cup \{(i, 0) \mid i \in V, \alpha_i > -\infty\} \cup \{(0, i) \mid i \in V, \beta_i < +\infty\}$. Then, using variables $(q_0, q) \in \mathbb{R} \times \mathbb{R}^V$, we can rewrite the problem as

$$\begin{aligned}
\text{minimize} \quad & \max_{i \in V_0} q_i - \min_{i \in V_0} q_i \\
\text{subject to} \quad & q_j - q_i \leq \hat{\gamma}_{ij} \quad \forall (i, j) \in E_0 \\
& q_0 = 0.
\end{aligned}$$

We further rewrite this problem as a linear programming problem with additional variables $q_s, q_t \in \mathbb{R}$ ($s, t \notin V_0$). Letting $q_s$ and $q_t$ represent $\max_{i \in V_0} q_i$ and $\min_{i \in V_0} q_i$, respectively, the objective

function is written as $q_s - q_t$, and this yields additional constraints $q_i - q_s \leq 0$ and $q_t - q_i \leq 0$ for $i \in V_0$. Thus, the negative of the above problem is written as

$$
\begin{array}{lll}
\text{maximize} & q_t - q_s & \\
\text{subject to} & q_j - q_i \leq \hat{\gamma}_{ij} & \forall (i,j) \in E_0 \\
& q_i - q_s \leq 0 & \forall i \in V_0 \\
& q_t - q_i \leq 0 & \forall i \in V_0 \\
& q_0 = 0.
\end{array} \tag{8}
$$

If we do not have the last constraint, $q_0 = 0$, (8) is the dual of the shortest $s$–$t$ path problem on a graph $\tilde{G} = (\tilde{V}, \tilde{E}, \tilde{w})$, where $\tilde{V} = V_0 \cup \{s, t\}$, $\tilde{E} = E_0 \cup \{(s, i) \mid i \in V_0\} \cup \{(i, t) \mid i \in V_0\}$, and

$$
\tilde{w}_{ij} = \begin{cases} \hat{\gamma}_{ij} & \text{for } (i,j) \in E_0 \\ 0 & \text{otherwise} \end{cases}
$$

for $(i, j) \in \tilde{E}$. Moreover, given any optimal solution $q' \in \mathbb{R}^{\tilde{V}}$ to (8) without the last constraint, $q^* = q' - q'_0 \mathbf{1}$ is also optimal and satisfies $q^*_0 = 0$. Hence, the remaining task is to solve the shortest path problem on $\tilde{G}$. Since $|\tilde{V}| = n + 3$, $|\tilde{E}| \leq m + 4n + 2$, and the $\mathrm{L}^\natural$-convexity of $S \neq \emptyset$ rules out the presence of negative cycles, the Bellman–Ford algorithm can solve the shortest path problem on $\tilde{G}$ in $\mathrm{O}(mn)$ time. More precisely, to obtain an optimal solution $q^* \in \mathbb{R}^{\tilde{V}}$ to (8), we find shortest paths from $s$ to all vertices in $\tilde{V} \setminus \{s\}$ with the Bellman–Ford algorithm, and we set the potential $q^*$ along the found paths so that $q^*_0 = 0$ holds. We obtain a desired projection as $p = \hat{p} + q^*_V$, where $q^*_V \in \mathbb{R}^V$ is the restriction of $q^* \in \mathbb{R}^{\tilde{V}}$ to $V$.

## E  Discussion on worst-case guarantees

While we have focused on prediction-dependent bounds, we can bound the worst-case time complexity of the DCA-based algorithms for bipartite matching and matroid intersection, as described below. The following bounds are, however, weaker than those of standard algorithms; therefore, we should run standard algorithms in parallel to obtain better worst-case guarantees, as discussed in Section 6.

As shown in [46, Theorem 4.17], the long-step steepest descent algorithm for an L-convex function $g : \mathbb{Z}^V \to Z \cup \{+\infty\}$ converges in $n \cdot \max\{g(p) - g(p + d) \mid p \in \mathbb{Z}^V, d \in \mathcal{N}_+\}$ iterations if the minimum minimizer $d \in \mathcal{N}_+$ is chosen in every local optimization. Thus, the DCA-based algorithm for bipartite matching (resp. matroid intersection) terminates in $\mathrm{O}(n^2)$ (resp. $\mathrm{O}(nr)$) iterations. Since a single iteration takes $\mathrm{O}(m\sqrt{n})$ (resp. $\mathrm{O}(\tau n r^{1.5})$) time, the total worst-case time complexity is $\mathrm{O}(mn^{2.5})$ (resp. $\mathrm{O}(\tau n^2 r^{2.5})$). We can also obtain a similar worst-case bound for discrete energy minimization if the derivatives of $\phi_i$ and $\psi_{ij}$ are bounded. We, however, do not know whether those bounds are tight; i.e., we do not know whether there is, for example, a worst-case bipartite-matching instance such that the DCA-based algorithm incurs $\Theta(mn^{2.5})$ time.

### E.1  Difficulty in recovering $\tilde{\mathrm{O}}(mn)$ time guarantee for bipartite matching

We also explain why the DCA-based algorithm cannot immediately recover the $\tilde{\mathrm{O}}(mn)$-time bound for bipartite matching ($\tilde{\mathrm{O}}$ hides logarithmic factors). In short, this is because the DCA-based algorithm is subtly different from the standard $\tilde{\mathrm{O}}(mn)$-time Hungarian method, as briefly described below (see [44, Section 18.5b] for more details).

Let $G = (L \cup R, E)$ be a bipartite graph with edge weights $w \in \mathbb{Z}^E$. Keeping a dual solution $p = (s, t) \in \mathbb{R}^{L \cup R}$, the DCA-based algorithm alternately computes a maximum matching of the tight subgraph $G^* = (L \cup R, E^*)$ with respect to $(s, t)$ from scratch using the Hopcroft–Karp algorithm and updates $(s, t)$. This results in the $\mathrm{O}(mn^{2.5})$ time complexity bound, as described above. By contrast, the standard Hungarian method keeps a maximum matching $M$ of $G^*$ in addition to a dual solution $(s, t)$. In every iteration, it augments $M$ or updates $(s, t)$ by searching an augmenting path in an orientation $D^*_M$ of $G^*$ defined from $M$. Moreover, the sequence of dual updates between two augmentations of $M$ can be aggregated into the single shortest-path searching on $D_M$, the orientation of $G$ by $M$ defined similarly to $D^*_M$, where edge lengths are given by $l_{ij} := s_i - t_j - w_{ij}$ for $ij \in E$. Since the edge lengths are non-negative, we can use Dijkstra's $\tilde{\mathrm{O}}(m)$-time algorithm to find a shortest path on $D_M$, thus achieving the $\tilde{\mathrm{O}}(mn)$ time.

To recover this $\tilde{O}(mn)$-time bound with the DCA-based algorithm, we need to convert it into the above standard Hungarian method. This modification, however, is specific to the bipartite-matching case and is not covered by the general DCA theory. Furthermore, the modification may worsen the prediction-dependent bound. This is because, while the DCA-based algorithm always updates a dual solution in every iteration, the standard Hungarian method may not when $M$ is augmented, implying that the bound based on $\|p^* - \hat{p}\|_\infty$ does not follow immediately.

## F   Experiments

We compare our DCA-based method with that of Dinitz et al. [17] using synthetic weighted bipartite matching instances. First, we summarize the differences between the two methods. Both methods are based on the same basic methodology: predict an (infeasible) dual solution $\hat{p}$, convert $\hat{p}$ into an initial feasible solution $p^\circ$, and warm-start the bipartite-matching solver with $p^\circ$. As for the weighted-bipartite-matching solver, both iteratively call the Hopcroft–Karp algorithm [26] to compute an optimal dual solution; hence the bipartite-matching solver is identical. The differences between our method and [17] lie in how to learn (infeasible) predictions $\hat{p}$ and how to obtain initial feasible solutions $p^\circ$. Below we detail these two differences.

**Learning predictions.**   Dinitz et al. [17] learned predictions $\hat{p}$ to minimize the empirical $\ell_1$-loss, while we have replaced the $\ell_1$-loss with the $\ell_\infty$-loss. In the following experiment, as described in Section 4 and [31], both methods learn predictions using the online gradient descent method (OGD), where the $\ell_1$- and $\ell_\infty$-losses are used for the method of [17] and ours, respectively.

**Obtaining feasible solutions.**   Dinitz et al. [17] converted an infeasible prediction $\hat{p}$ into feasible $p^\circ$ using a greedy approximation algorithm tailored to the $\ell_1$-loss. By contrast, we obtain feasible solutions $p^\circ$ by minimally shifting $\hat{p}$ in the all-one direction as in Section 3.1 and rounding the resulting vector to the nearest integer point.

Due to the above differences, the method of [17] and ours yield different initial feasible solutions $p^\circ$, which are fed to the common bipartite-matching solver. The following experiment examines how this difference affects the number of iterations taken by the bipartite-matching solver.

**Settings.**   We generated three random bipartite graphs $(V, E)$ such that $V = L \cup R$ and $|L| = |R| = 5$ (i.e., $n = 10$) with three probability values $\theta = 0.5, 0.7, 0.9$ for edge creation. For each graph, we generated $T = 1000$ sets $\{w_e\}_{e \in E}$ of random edge weights by setting $w_e = 5 + \lfloor u \rceil$ for each $e \in E$, where $u$ is drawn from the standard normal distribution. We thus created 1000 bipartite-matching instances with various edge weights for each graph. For each sequence of the 1000 instances, we iteratively predicted a dual solution $\hat{p}$ with OGD, converted $\hat{p}$ into a feasible solution $p^\circ$, and solved the instance by warm-starting the bipartite-matching solver with $p^\circ$. We implemented OGD based on that of [45, Section 2.4]. The original OGD uses the step size of $\frac{B}{L\sqrt{2T}}$, where $B$ is the radius of the $\ell_2$-ball that contains optimal predictions and $L$ is the Lipschitz constant of the loss functions. In our setting, we have $B = C\sqrt{n} = n^{1.5}\|w\|_\infty$, where we set $\|w\|_\infty = 9$ since all the instances satisfied $|w_e| \le 9$ for all $e \in E$. The Lipschitz constant $L$ was set to 1 for the $\ell_\infty$-loss and $\sqrt{n}$ for the $\ell_1$-loss. We, however, observed that the step size of $\frac{B}{L\sqrt{2T}}$ was too large to achieve better performances than a baseline method with random initial solutions $p^\circ$, which we call the cold-start baseline. Thus, we rescaled the step size as $\alpha \times \frac{B}{L\sqrt{2T}}$, where $\alpha$ is the scaling parameter. We let $\alpha = 0.001, 0.01, 0.1$.

**Results.**   Figure 2 presents the cumulative number of iterations taken by the common weighted-bipartite-matching solver for ours ($\ell_\infty$), the method of [17] ($\ell_1$), and the cold-start baseline (Cold). We conducted 10 independent trials with random initial points of OGD drawn from the $n$-dimensional standard normal distribution; the error band indicates the 95% confidence interval of the 10 random trials. Both ours and the method of [17] took fewer iterations than the cold-start baseline, and ours with $\alpha = 0.01, 0.1$ outperformed the method of [17]. The results suggest the practical usefulness of our DCA-based framework.

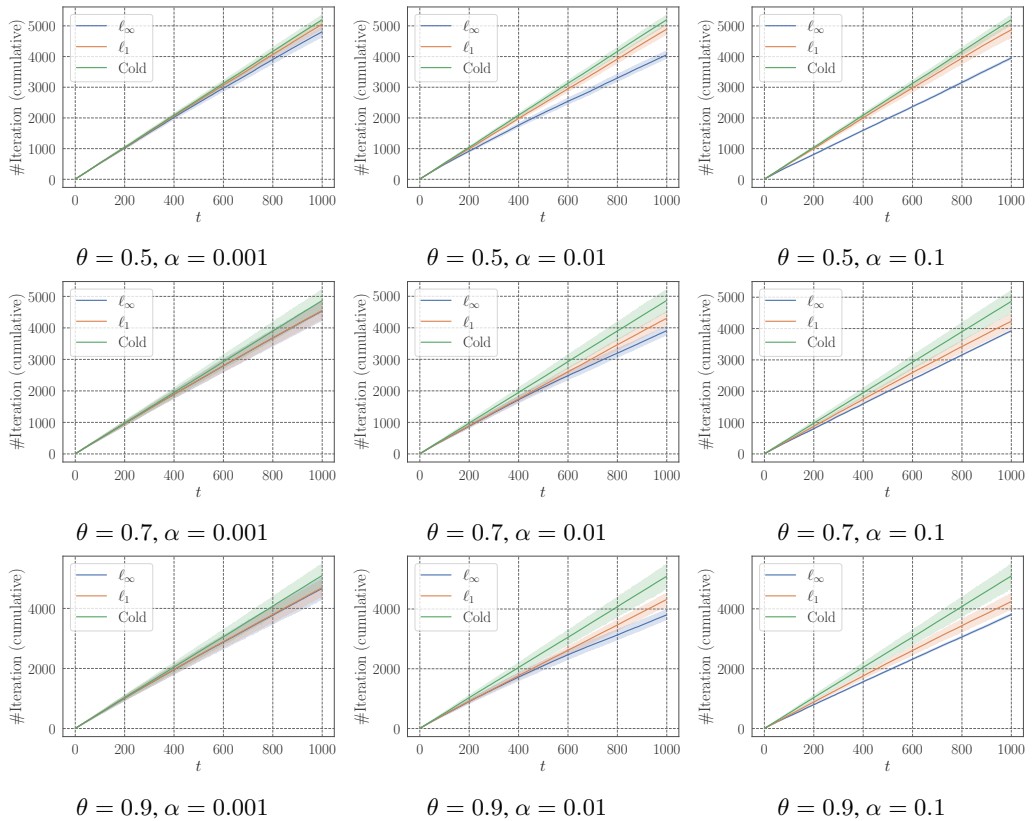

Figure 2: The cumulative number of iterations taken by the bipartite-matching solver for each method. The parameter $\theta$ is the probability of edge creation, and $\alpha$ is the scaling parameter to control the step size of OGD. The error band indicates the $95\%$ confidence interval over the 10 random trials.