# OpenReview forum: "Discrete-Convex-Analysis-Based Framework for Warm-Starting Algorithms with Predictions"
_NeurIPS.cc/2022/Conference — NeurIPS 2022 Accept_

### Official Review · Reviewer_KW47 · 2022-07-02

**Rating:** 6
**Confidence:** 1
**Soundness:** 3 good
**Presentation:** 3 good
**Contribution:** 3 good

**Summary:**

This paper presents results illustrating the possible benefits of initializing learning algorithms based on predictions. By using a discrete convex analysis framework, the results indicate improved bounds for the sample and time complexity. The usefulness of the framework is shown by applications to weighted perfect bipartite matching, weighted matroid intersection, and discrete energy minimization. Finally, whereas the choice whether to learn primal or dual solutions has typically been made somewhat heuristically, the paper uses the proposed framework to give principled guidelines.

**Questions:**

What is the reason that your approach is unable to recover optimal worst-case bounds? Is this inherent to the approach, or can it be alleviated? (I understand that this is not crucial, as discussed in the end of the paper)
You discuss in Section 6 that the issue above can be mitigated by running a standard algorithm in parallel. It seems as if the time complexity of such an approach would, compared to alternatives, never be worse by more than a constant factor 2. Is this a correct interpretation? What is the worst case?
Can similar lower bounds be shown for the time complexity?

Minor things:
line 28: worse->worst
line 113: "the the"

**Limitations:**

The authors clearly state the limitations, but an extended discussion on what classes of problems enjoy the different convexity notions could be beneficial.

**Strengths And Weaknesses:**

The paper is clearly written and structured, addresses a pertinent issue, and seems to provide novel improvements on prior work. The presentation is neat.
The improvements over prior results are significant; for instance, the time complexity of some algorithms is improved by up to a factor $n$, where $n$ is the number of vertices.

One weakness of the results is that, unlike many previous analyses where the possible performance of learning algorithms due to the use of predictions, the analysis is unable to recover optimal worst-case bounds.
A more extensive discussion of the applicability of the given convexity assumptions and how these contrast with other notions of convexity could be beneficial in order to more clearly motivate the work and clarify the benefits and limitations.

---

> ### Author Response · Authors · 2022-08-02
> **Response to Reviewer KW47**
>
> We appreciate the reviewer's thoughtful comments and questions on the worst-case bounds.
>
> &nbsp;
> > What is the reason that your approach is unable to recover optimal worst-case bounds? Is this inherent to the approach, or can it be alleviated?
>
> Our approach based on discrete convex analysis (DCA) can sometimes enjoy worst-case bounds, which, however, require instance-specific discussions and are often weaker than those of the standard algorithms without warm-starts (hence we should resort to the parallel execution in practice, as described in Section 6). Since our main focus is to develop a general framework for warm-starting algorithms with predictions, we did not go into the details of the worst-case bounds to avoid complications. In response to subsequent questions, we will detail the worst-case bounds of our DCA-based approach for bipartite matching, matroid intersection, and discrete energy minimization.
>
> We agree that an additional mention of the worst-case bounds and the applicability of our DCA framework will be instructive to the readers. We will add discussions on them to the final version if accepted.
>
> &nbsp;
> >You discuss in Section 6 that the issue above can be mitigated by running a standard algorithm in parallel. It seems as if the time complexity of such an approach would, compared to alternatives, never be worse by more than a constant factor 2. Is this a correct interpretation?
>
> The interpretation is correct. Please note that we can independently execute our algorithm with warm-starts and a standard algorithm. Therefore, if two computational environments are available, the parallel execution takes at most the same time as the standard algorithm.
>
> &nbsp;
> >What is the worst case? Can similar lower bounds be shown for the time complexity?
>
> We can derive the worst-case bounds for bipartite matching and matroid intersection as follows, although they are weaker than those of standard algorithms.
>
> It is known that the long-step steepest descent algorithm for an L-convex function $g: \mathbb{Z}^V \to \mathbb{Z} \cup \\{+\infty\\}$ converges in $n\max\\{g(p)-g(p+d):p\in\mathbb{Z}^V,d\in\mathcal{N}\_+\\}$ iterations if the minimum minimizer $d\in\mathcal{N}\_+$ is chosen in every local optimization [46, Theorem 4.17]. Thus, our warm-starting algorithm for bipartite matching (resp. matroid intersection) terminates in $O(n^2)$ (resp. $O(nr)$) iterations. Since a single iteration takes $O(m\sqrt{n})$ (resp. $O(\tau nr^{1.5})$) time, the total worst-case time complexity is $O(mn^{2.5})$ (resp. $O(\tau n^2 r^{2.5})$).
>
> We can also obtain a similar worst-case bound for discrete energy minimization when the derivatives of $\phi_i$ and $\psi_{ij}$ are bounded. We, however, do not know whether those bounds are tight, i.e., whether there is a worst-case instance of bipartite matching, for example, such that our algorithm incurs $\Theta(mn^{2.5})$ time.
>
> &nbsp;
> ### On bipartite matching
> We would like to mention why our DCA approach cannot immediately recover the $\tilde{O}(mn)$-time bound for bipartite matching ($\tilde{O}$ hides logarithmic factors).
> In a nutshell, this is because our algorithm is subtly different from the standard $\tilde{O}(mn)$-time Hungarian method as follows (see [44, Section 18.5b] for more details).
>
> Let $G=(L\cup R,E)$ be a bipartite graph with edge weights $w\in\mathbb{Z}^E$. Keeping a dual solution $p = (s,t)\in\mathbb{R}^{L\cup R}$, our DCA-based algorithm alternately computes a maximum matching of the tight subgraph $G^*=(L\cup R,E^*)$ with respect to $(s,t)$ from scratch with the Hopcroft--Karp algorithm and updates $(s,t)$. This results in the $O(mn^{2.5})$ time complexity bound, as discussed above.
> By contrast, the standard Hungarian method keeps a maximum matching $M$ of $G^*$ and a dual solution $(s,t)$.
> In every iteration, it augments $M$ or updates $(s,t)$ by searching an augmenting path in an orientation $D^\*\_M$ of $G^\*$ defined from $M$. Moreover, the sequence of dual updates between two augmentations of $M$ can be aggregated into the single shortest-path searching on $D_M$, the orientation of $G$ by $M$ defined similarly to $D_M^*$, with respect to edge length $l_{ij}\coloneqq s_i-t_j-w_{ij}$ for $ij\in E$. Since the edge lengths are non-negative, we can use Dijkstra's $\tilde{O}(m)$-time algorithm, thus achieving the $\tilde{O}(mn)$ time.
>
> Since the above difference is too involved, we omitted the details in the manuscript.
> To recover the $\tilde{O}(mn)$-time bound, we need to convert our algorithm into the above standard Hungarian method.
> This modification, however, is specific to bipartite matching and is not covered by the general DCA theory.
> Moreover, the modification may worsen the prediction-dependent bound. This is because, while our DCA-based algorithm always updates the dual solution in every iteration, the standard Hungarian method does not when $M$ is augmented, implying that the bound based on $\\|p^*-\hat p\\|_\infty$ does not follow immediately.

---

### Official Review · Reviewer_TRn1 · 2022-07-11

**Rating:** 6
**Confidence:** 4
**Soundness:** 3 good
**Presentation:** 3 good
**Contribution:** 3 good

**Summary:**

This paper extends recent work by Dinitz et al. on using the learning-augmented algorithms model to speed up graph algorithms by warm-starting the dual solutions (in particular, they focused on the Hungarian algorithm for minimum-cost bipartite matching). In this paper, the authors interpret this idea of warm-starting in the context of discrete convex analysis, inspired by the idea that warm-starting is a well studied concept in convex optimization. Through this lens, they give an improvement in the min-cost matching problem, reducing the dependence on the prediction error from linear in the $\ell_1$ error to linear in the $\ell_\infty$ error. Their framework is rather general and the authors apply it to get learning-augmented algorithms and bounds for weighted matroid intersection and discrete energy minimization as well. The authors also provide PAC bounds on learning predictions for these problems under the $\ell_\infty$ norm.

**Questions:**

Do you think that the DCA-based algorithms actually improve on the work of Dinitz et al in practice, or is it rather the case that you can give a tighter analysis of these algorithm than they could?

**Limitations:**

I found the discussion of limitations to be adequate. I do not foresee any specific negative impacts of this work as it is quite theoretical and abstracted away from applications.

**Strengths And Weaknesses:**

Strengths
- The connection between warm-starting in Dinitz et al and in the framework of discrete convex analysis is interesting and general.
- The improvements in the bounds from prior work are significant, changing the error dependence from $\ell_1$ to $\ell_\infty$.
- I thought the discussion of learning primal vs. dual solutions in Section 5 was very interesting. It is worthwhile to consider what factors make some predictions useful or not for algorithms with predictions.

Weaknesses
- For large graphs, the PAC bounds require many samples. In particular, if we set $C=n\|w\|_\infty$, we need at least $\frac{n^2 \|w\|_\infty^2}{\epsilon^2}$ sample instances which for large graphs seems extreme. The bounds are a significant improvement over the bounds given in prior work but I'm not sure how informative they actually are.
- It seems that many of the techniques used in this paper come almost directly from prior work. Nonetheless, I think the application and combination of these ideas makes for some interesting insights and results on learning-augmented algorithms.
- Perhaps most importantly, seeing as both this and the prior work are in some sense not comparing themselves to the best theoretical algorithms for matching (even if the predictions are near-perfect, the theoretical bounds will be worse), it would be more compelling if some experiments were done to show that the this discrete convex analysis approach is actually a better way to approach the problem than Dinitz et al.

---

> ### Author Response · Authors · 2022-08-02
> **Response to Reviewer TRn1**
>
> We are grateful to the reviewer for providing valuable comments and questions.
>
> > Do you think that the DCA-based algorithms actually improve on the work of Dinitz et al in practice, or is it rather the case that you can give a tighter analysis of these algorithm than they could?
>
> We think our practical improvement from Dinitz et al. is not very large. Indeed, reading the comment, we conducted a preliminary experiment to compare the two methods and observed that their performances are similar. We will detail the experiment later.
>
> Theoretically, however, we believe our work has significantly improved Dinitz et al. in two aspects.
>
> 1. As mentioned in the comment, our bound depending on the $\ell_\infty$-distance offers a tighter result than the previous $\ell_1$-dependent one, which yields up to $n$ times improvement of the time complexity bound.
>
> 2. More importantly, our discrete-convex-analysis-based framework has revealed a broad class of discrete optimization problems (i.e., $\text{L}$/$\text{L}^\natural$-convex minimization problems) that accepts a geometric understanding of why learning of warm-starts is effective.
>
> In summary, our main improvement from Dinitz et al. is the theoretical results, i.e., tighter analysis and extension of the class of problems.
>
> &nbsp;
> ## Preliminary experiment
> We here present preliminary experimental results. We will work more on the experiments and, if accepted, add the results to the supplementary.
>
> First, we summarize the differences between Dinitz et al. and ours. Once a feasible dual solution $p^\circ \in \mathbb{R}^V$ is given, both methods solve a bipartite matching instance by iteratively calling the Hopcroft-Karp algorithm. Thus, the bipartite-matching solvers used in Dinitz et al. and our DCA-based framework are identical. The differences between Dinitz et al. and our method lie in how to learn (infeasible) predictions $\hat p$ and obtain feasible solutions $p^\circ$.
>
> - **Learning predictions.** Dinitz et al. learned predictions to minimize the empirical $\ell_1$-loss, while we have replaced the $\ell_1$-loss with the $\ell_\infty$-loss. In the following experiment, we learn predictions using the online gradient descent method (OGD), as described in Section 4 and [31], where the loss functions are $\ell_1$- and $\ell_\infty$-losses for Dinitz et al. and ours, respectively.
>
> - **Obtaining feasible solutions.** Dinitz et al. converted an infeasible prediction $\hat p$ into a feasible initial solution $p^\circ$ with a greedy approximation algorithm tailored to the $\ell_1$-loss. By contrast, we obtain feasible solutions $p^\circ$ by minimally shifting $\hat p$ in the all-one direction as in lines 198--200 and rounding the resulting vector to the nearest integer point.
>
> Due to the above differences, the method of Dinitz et al. and ours yield different feasible initial solutions $p^\circ$, which are fed to the common bipartite-matching solver.
> The following experiment examines how this difference affects the number of iterations taken by the bipartite-matching solver.
>
> ### Settings
> We generated a random bipartite graph $(V, E)$ such that $V = L \cup R$, $|L| = |R| = 5$, and each $(i, j) \in L \times R$ has an edge with a probability of $0.8$.
> Fixing this graph, we generated $1000$ sets $\\{w_e\\}_{e\in E}$ of random edge weights by setting $w_e = 5 + \lfloor u \rceil$ for each $e \in E$, where $u$ is drawn from the standard normal distribution. We thus created $1000$ bipartite-matching instances with various edge weights. For each of these instances, we iteratively predicted a dual solution $\hat p$ with OGD, converted $\hat p$ into a feasible solution $p^\circ$, and solved the instance by warm-starting the bipartite-matching solver with $p^\circ$.
>
> ### Results
> We solved the $1000$ instances using the method of Dinitz et al. and ours and computed the average number of iterations for each method.
> Our method took $3.458$ iterations on average, while the method of Dinitz et al. took $3.614$ iterations.
> We also solved the $1000$ instances using the solver without warm-starts and observed that it took $5.122$ iterations on average.
> To conclude, both methods successfully took advantage of learned predictions, and their empirical performances were similar.

---

> > ### Comment · Reviewer_TRn1 · 2022-08-07
> > **Reply to author comments**
> >
> > Thanks for your response and for running this experiment! I think it helps to see the contribution in this work as giving an extended and tighter analysis of this warm-starting approach.

---

### Official Review · Reviewer_zvz6 · 2022-07-13

**Rating:** 7
**Confidence:** 3
**Soundness:** 4 excellent
**Presentation:** 3 good
**Contribution:** 3 good

**Summary:**

This paper considers using predictions to improve algorithm running time via warm start as in the recent work due to Dinitz et al. 2021.  The authors propose a framework based on discrete convex analysis which applies a discrete steepest decent approach to go from an initial feasible solution to a global optimum.  Each iteration requires solving a local optimization problem to find a descent direction and the number of iterations can be bounded by $O(||p-p^*||_{\infty})$, where $p$ is the initial solution and $p^*$ is an optimal solution.  Algorithms for projecting infeasible predicted solutions onto the feasible set (with respect to the $\ell_\infty$) norm and efficiently learning an initial predicted solution are also given to complete the framework.

Using this framework the authors give warm-start-with-predictions algorithms for:
- weighted bipartite matching (improving the result of Dinitz et al. 2021)
- weighted matroid intersection
- Discrete energy minimization
- The more general class of L/L♮-convex functions.



**Questions:**

- How does the proposed approach for weighted bipartite matching compare experimentally to Dinitz et al. 2021?

**Strengths And Weaknesses:**

This paper gives a nice improvement over Dinitz et al 2021 for weighted bipartite matching and also extends the ideas to other problems.  The writing is a bit dense, but overall well explained.  An experimental evaluation may be interesting, but is probably not necessary for the current submission.

---

> ### Author Response · Authors · 2022-08-02
> **Response to Reviewer zvz6**
>
> We appreciate the reviewer's careful reading and thoughtful comments.
>
> > How does the proposed approach for weighted bipartite matching compare experimentally to Dinitz et al. 2021?
>
> We conducted a preliminary experiment on synthetic data. We observed that the method of Dinitz et al. and ours exhibited similar performances, and both outperformed the baseline without warm-starts. For details of the experiment, please see the [response to Reviewer TRn1](https://openreview.net/forum?id=-GgDBzwZ-e7&noteId=gmoXgeiHOM).
>
> &nbsp;
> ### Minor comment
> Perhaps the "Ethics Flag" is marked "Yes" mistakenly. Please change it to "No" if you have no ethical concerns.

---

### Meta-Review · Area_Chair_ScF3 · 2022-08-23

**Recommendation:** Accept
**Confidence:** Certain

**Metareview:**

In this paper, the authors provide new theoretical guarantees for augmenting algorithms with learned predictions. Based on discrete convex analysis (DCA), they generalize previous results of Dinitz et al, and obtain better time complexity bounds for a number of online problems. The application of DCA to online algorithms with predictions is interesting, and the improvements in the bounds are significant.

**Award:**

No

---

### Decision · Program_Chairs · 2022-09-14

Accept